# Actin-regulated Siglec-1 nanoclustering influences HIV-1 capture and virus-containing compartment formation in dendritic cells

Enric Gutiérrez-Martínez[1], Susana Benet Garrabé[2], Nicolas Mateos[1], Itziar Erkizia[3], Jon Ander Nieto-Garai[4], Maier Lorizate[4], Kyra JE Borgman[1], Carlo Manzo[5], Felix Campelo[1], Nuria Izquierdo-Useros[3,6,7]*, Javier Martinez-Picado[3,6,7,8,9]*, Maria F Garcia-Parajo[1,9]*

[1]ICFO – Institut de Ciencies Fotoniques, The Barcelona Institute of Science and Technology, Barcelona, Spain; [2]Lluita contra la sida foundation, Infectious Diseases Department, Hospital Germans Trias i Pujol, Badalona, Spain; [3]IrsiCaixa AIDS Research Institute, Badalona, Spain; [4]Department of Biochemistry and Molecular Biology, Universidad del País Vasco, Bilbao, Spain; [5]Facultat de Ciències i Tecnologia, Universitat de Vic - Universitat Central de Catalunya, Vic, Spain; [6]Germans Trias i Pujol Research Institute (IGTP), Badalona, Spain; [7]CIBER de enfermedades infecciosas, Madrid, Spain; [8]AIDS and Related Illnesses, Centre for Health and Social Care Research (CESS), Faculty of Medicine, University of Vic - Central University of Catalonia, Vic, Spain; [9]ICREA, Barcelona, Spain

*For correspondence:
nizquierdo@irsicaixa.es (NI-U);
jmpicado@irsicaixa.es (JM-P);
maria.garcia-parajo@icfo.eu
(MFG-P)

**Abstract** The immunoglobulin-like lectin receptor CD169 (Siglec-1) mediates the capture of HIV-1 by activated dendritic cells (DCs) through binding to sialylated ligands. These interactions result in a more efficient virus capture as compared to resting DCs, although the underlying mechanisms are poorly understood. Using a combination of super-resolution microscopy, single-particle tracking and biochemical perturbations we studied the nanoscale organization of Siglec-1 on activated DCs and its impact on viral capture and its trafficking to a single viral-containing compartment. We found that activation of DCs leads to Siglec-1 basal nanoclustering at specific plasma membrane regions where receptor diffusion is constrained by Rho-ROCK activation and formin-dependent actin polymerization. Using liposomes with varying ganglioside concentrations, we further demonstrate that Siglec-1 nanoclustering enhances the receptor avidity to limiting concentrations of gangliosides carrying sialic ligands. Binding to either HIV-1 particles or ganglioside-bearing liposomes lead to enhanced Siglec-1 nanoclustering and global actin rearrangements characterized by a drop in RhoA activity, facilitating the final accumulation of viral particles in a single sac-like compartment. Overall, our work provides new insights on the role of the actin machinery of activated DCs in regulating the formation of basal Siglec-1 nanoclustering, being decisive for the capture and actin-dependent trafficking of HIV-1 into the virus-containing compartment.

## Editor's evaluation

Siglec-1 (CD169), a plasma membrane-associated sialic acid-binding lectin, has been implicated in the capture of HIV and other viruses by dendritic cells and macrophages, however, the molecular details of how HIV particles are captured by Siglec-1 are poorly understood. In this important paper, the authors use super-resolution imaging methods to analyse the cell surface distribution of Siglec-1

on immature and mature dendritic cells to study the regulation of Siglec-1 distribution by actin and regulators of actin polymerization and to understand how virus-Siglec-1 engagement leads to virus sequestration within so-called virus containing compartments. This compelling study has relevance for researchers studying the engagement of HIV and many other viruses with cells, as well as researchers interested in the mechanisms regulating receptor distribution and function on cells.

## Introduction

Dendritic cells (DCs) are a specialized group of leukocytes that play an essential role in the innate and adaptive immunity through their function as antigen presenting cells (*Banchereau and Steinman, 1998*). DCs patrol peripheral tissues capturing invading pathogens, including viruses such as HIV-1, which are then degraded into antigens as the cells migrate toward the lymph nodes to activate T-cell responses. However, HIV-1 can also exploit DCs as vehicles to *trans*-infect CD4$^+$ T cells, a process that allows the dissemination of the virus through cell-to-cell contacts (*Cameron et al., 1992b*; *Pope et al., 1994*). Although iDCs express receptors and co-receptors required for HIV-1 infection (*Granelli-Piperno et al., 1996*; *Turville et al., 2002*), their infection rates in culture are lower than the ones for activated CD4$^+$ T cells or macrophages (*Cameron et al., 1992a*; *Granelli-Piperno et al., 1998*; *Granelli-Piperno et al., 1999*; *Pope et al., 1995*). Moreover, iDCs direct part of the trapped viruses to the endolysosomal degradation pathway, which further decreases both the susceptibility of iDCs to infection and the load of viral particles that reach the cytoplasm to initiate their intracellular replication (*Turville et al., 2004*). As such, iDCs are inefficient *trans*-infecting T cells. In strong contrast, activation of DCs by lipopolysaccharide (LPS) or interferon (IFN), which are immunomodulatory signals present during HIV-1 infection, diminishes their susceptibility to infection but significantly increases the capture of HIV-1 and enhances the *trans*-infection of CD4$^+$ T cells (*Izquierdo-Useros et al., 2007*; *Wang et al., 2007*).

The immunoglobulin-like lectin receptor CD169 (Siglec-1) is a key molecule in the HIV-1 capture and *trans*-infection of T cells by activated DCs (*Izquierdo-Useros et al., 2012b*; *Puryear et al., 2013*). Siglec-1 is a type I transmembrane protein that contains an amino-terminal V-set domain, which can directly bind to sialylated ligands like the ones present in gangliosides of the HIV-1 membrane (*Crocker et al., 2007*). In LPS-activated DCs (i.e., mature DCs, mDCs), once HIV-1 engages Siglec-1, a large amount of viral particles accumulate in a compartment connected to the extracellular milieu and enriched in several tetraspanin proteins (*Garcia et al., 2005*; *Hyun et al., 2008*; *Izquierdo-Useros et al., 2011*; *Izquierdo-Useros et al., 2012b*; *Puryear et al., 2013*). The formation of this virus-containing compartment (VCC) reduces HIV-1 endolysosomal degradation while the accumulation of viruses enhances the viral load to *trans*-infect T cells (*Izquierdo-Useros et al., 2012b*).

Activation of DCs by LPS or IFNα upregulates the expression levels of Siglec-1, which partially explains their higher HIV-1 capture ability as compared to iDCs (*Akiyama et al., 2015*; *Izquierdo-Useros et al., 2012b*). However, multiple studies indicate that the spatial organization of plasma membrane receptors in immune cells as well as structural characteristics of their ligands also play an important role in the regulation of ligand/receptor interactions (*Cambi et al., 2004*; *de Bakker et al., 2007*; *Gold and Reth, 2019*; *Neumann et al., 2019*; *Sherman et al., 2011*; *Torreno-Pina et al., 2016*; *Torreno-Pina et al., 2014*). For instance, the C-type lectin DC-specific intercellular adhesion molecule grabbing non-integrin (DC-SIGN), which also mediates the capture and internalization of several pathogens, including HIV-1 (*Kwon et al., 2002*), organizes in small nanoclusters in the plasma membrane of iDCs (*Cambi et al., 2004*; *de Bakker et al., 2007*). Such confined distribution increases the avidity of DC-SIGN for multimeric ligands by providing docking nano-platforms particularly efficient for the capture, internalization and degradation of viral particles (*Cambi et al., 2004*). Recently, another study established a link between the physical properties of synthetic glycopolymers carrying the mannose ligands for DC-SIGN and the internalization rates of the receptor (*Jarvis et al., 2019*). Longer polymers with a higher number of mannose copies internalize faster than shorter ones, and polymer aggregation in the surface of particulate antigens diverge DC-SIGN receptors from endocytosis, directing them to invaginated pockets similar to the VCC harboring HIV-1 (*Jarvis et al., 2019*). In the context of Siglec-1, artificial liposomes carrying the ganglioside GM4, which contains a sialic acid moiety bound to a single galactose, are barely captured by mDCs (*Izquierdo-Useros et al., 2012a*). In contrast, liposomes with GM1, GM2, or GM3, whose sialic acid is bound to a lactose head

group, efficiently bind to Siglec-1 with different concentration-dependent rates (*Izquierdo-Useros et al., 2012a*; *Puryear et al., 2012*). Overall, these findings indicate that the final outcome of viral capture and intracellular trafficking is the combined result of, on the one hand, expression levels and nanoscale organization of the receptors and, on the other hand, concentration, spatial distribution, and intrinsic structural properties of the ligands.

Here, we used a combination of super-resolution stimulated emission depletion (STED) microscopy and single-particle tracking (SPT) approaches to unravel the role of Siglec-1 nanoscale organization in the capture and trafficking of HIV-1 toward the VCC on activated DCs. Overall, our work reveals that distinctive components of the activated DC actin machinery regulate in a synergistic fashion the different stages of viral capture, trafficking and VCC formation, providing activated DCs with an increased capacity to *trans*-infect T cells.

## Results

### DC activation induces the formation of Siglec-1 nanoclusters with decreased mobility

The *trans*-infection capacity of DCs correlates with the expression levels of Siglec-1, which are increased upon cell activation with LPS or IFN (*Izquierdo-Useros et al., 2012b*; *Puryear et al., 2013*). However, the spatial organization of Siglec-1 receptors in resting or activated DCs is not yet known. To address whether DC maturation alters Siglec-1 distribution in the plasma membrane, we first examined the nanoscale organization of Siglec-1 by STED microscopy in iDCs and LPS-treated DCs (mDCs) differentiated from peripheral blood monocytes (PBMCs) (*Figure 1A*). With a lateral resolution of ~80 nm (*Figure 1—figure supplement 1A*), we discriminated individual Siglec-1 fluorescent spots on the cell surface and measured their peak intensities. To quantify the number of Siglec-1 molecules per spot in the plasma membrane of iDC and mDC we relied on the intensity obtained from individual antibodies (Abs) on glass, corresponding to single molecules (*Figure 1—figure supplement 1B–D* and *Figure 1—source data 1*; *Martínez-Muñoz et al., 2018*). LPS-mediated DC maturation induced a higher fraction (~57% vs. ~37%) of Siglec-1 dimers and small nanoclusters with >3 molecules/spot as compared to iDCs, where Siglec-1 was mainly found as monomers (~57% vs. ~22%) (*Figure 1A, B* and *Figure 1—source data 1*). The increase in the average number of molecules per spot also coincided with an average increase in spot sizes (*Figure 1C* and *Figure 1—source data 1*) and a significant increase in the proximity of Siglec-1 spots on mDCs as compared to iDCs (*Figure 1D* and *Figure 1—source data 1*). To exclude any potential artifacts associated to the labeling strategy and/ or fixation protocol, we performed STED imaging using single-chain Abs (instead of full Abs) on mDCs fixed either with 4% paraformaldehyde (PFA) or with 1% PFA + 0.2% glutaraldehyde (GA). Analysis of multiple STED images using single-chain Abs showed no difference in terms of nanoclustering capacity of Siglec-1 on mDCs as compared to full Ab labeling, regardless of the fixation method (*Figure 1—figure supplement 1E–I*, *Figure 1—figure supplement 1—source data 1*, *Figure 1— figure supplement 1—source data 2*). Thus, these controls safely exclude any potential perturbation induced by the Abs. or cell fixation on the nanoclustering degree of Siglec-1.

To rule out that nanoclustering and their spatial proximity were simply a consequence of the increased expression of Siglec-1 upon DC maturation, we performed simulations of spots randomly distributed through the cell area using identical Siglec-1 molecular densities as obtained from the experimental data (*Figure 1—figure supplement 1J*). Interestingly, normalization of the experimental data to the randomized values preserved the significant differences between mDC and iDC in spot size, proximity between spots and number of molecules per spot (*Figure 1—figure supplement 1K–M* and *Figure 1—figure supplement 1—source data 2*). Thus, these results indicate the existence of real, active basal nanoclustering of Siglec-1 after DC maturation.

The differences in Siglec-1 nanoscale organization between iDC and mDC might result from a difference in receptor mobility in the plasma membrane (*Martínez-Muñoz et al., 2018*; *Torreno-Pina et al., 2016*). To characterize the mobility of Siglec-1 we performed single-particle tracking (SPT) of individual Siglec-1 molecules labeled at low density using an anti-mouse Fab fragment conjugated to Atto488. Albeit different types of mobility were observed on both cell types (representative tracks in *Figure 1E*), DC maturation led to an overall reduction of Siglec-1 diffusion (*Figure 1F*) and increased the fraction of immobile receptors (diffusion values <0.002 $\mu m^2$/s), as compared to iDCs (~28% vs.

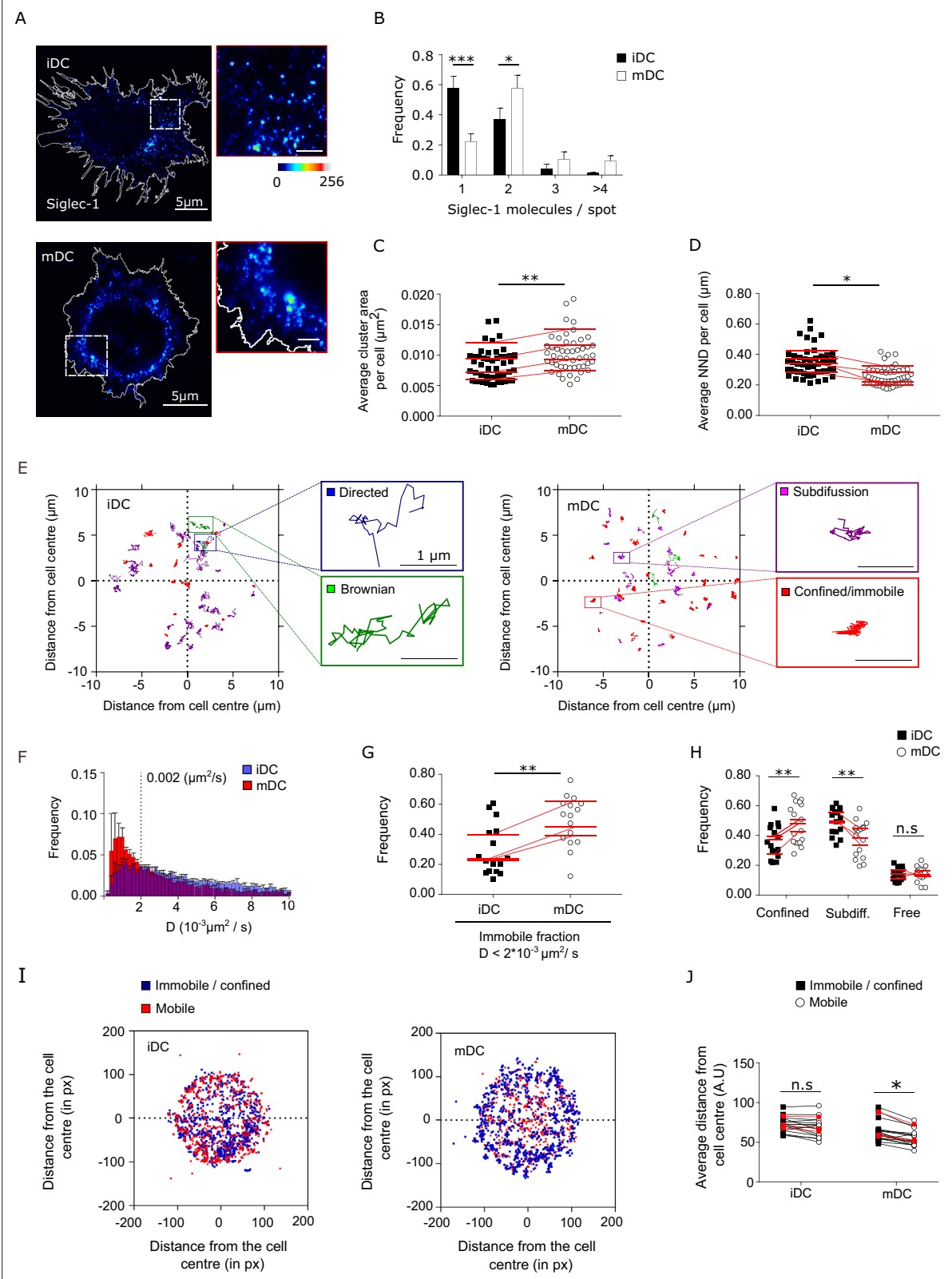

**Figure 1.** Dendritic cell (DC) activation induces the formation of Siglec-1 nanoclusters with decreased mobility. (**A**) Representative STED images of Siglec-1 in immature DCs (iDCs) and mature DCs (mDCs). The pseudo-color code denotes the intensity of individual Siglec-1-labeled spots, from monomer (dark blue) to nanocluster (red-to-white). The insets show enlarged regions highlighted by the white boxes on the main images (scale bar: 1 µm). Siglec-1 has been labeled using full Abs (see Materials and methods). Control experiments using single-chain Abs and different fixation methods

*Figure 1 continued on next page*

*Figure 1 continued*

are shown in *Figure 1—figure supplement 1E–I*, *Figure 1—figure supplement 1—source data 1*, and *Figure 1—figure supplement 1—source data 2*. (**B**) Frequency of the number of Siglec-1 molecules per spot, in iDC and mDC. Bars represent the mean ± standard error of the mean (SEM) of 3 different donors (minimum of 9 cells/donor and condition). (**C**) Average size of Siglec-1 spots and (**D**) proximity between individual spots, calculated by measuring the nearest neighbor distance (NND) between spots per cell, in iDC and mDC. Each symbol in (**C, D**) corresponds to an individual cell, red lines are the average value on iDCs and mDCs for each donor (4 donors, 9 cells/donor and cell type). (**E**) Representative Siglec-1 trajectories with the receptor labeled with single-chain Abs at sub-labeling conditions as recorded on an iDC (left) and an mDC (right). The magnified insets show examples of different types of motion as classified by the MSS analysis. (**F**) Frequency of the diffusion coefficients for individual Siglec-1 trajectories on iDCs and mDCs. The dash vertical line corresponds to the diffusion threshold to separate immobile from mobile particles ($0.002\ \mu m^2$/s). Each data set represents the mean ± SEM of 3 donors (minimum of 3 cells and 83 trajectories/cell). (**G**) Fraction of immobile trajectories ($<0.002\ \mu m^2$/s). Each symbol corresponds to an individual cell, red lines are the average value on iDCs and mDCs for each donor (3 donors). (**H**) Fraction of mobile trajectories ($>0.002\ \mu m^2$/s) classified as confined, sub-diffusive or free. Each symbol corresponds to an individual cell, red lines show the average value on iDCs and mDCs for 3 donors analyzed. (**I**) Plots showing the center of mass of individual Siglec-1 trajectories (average $x,y$ position in all the frames of a given trajectory) in iDC and mDC. Blue dots correspond to immobile and confined trajectories, and red dots correspond to sub-diffusive and free trajectories. The graph shows all the trajectories analyzed for a minimum of 8 cells per condition from one donor. (**J**) Distance from the cell center of immobile/confined (black squares) and sub-diffusive/free (empty circles) trajectories in iDC and mDC. Black squares are paired to the empty circles within the same cell. Red symbols correspond to the average values of a minimum of 3 iDC and mDC cells per donor (3 donors). ns, p > R 0.05, * p < % 0.05, ** p < % 0.001; *** p < % 0.0001.

The online version of this article includes the following source data and figure supplement(s) for figure 1:

**Source data 1.** Excel file containing the source data for *Figure 1B–D, G, H*.

**Figure supplement 1.** Different nanoscale distribution of Siglec-1 in immature versus mature DCs.

**Figure supplement 1—source data 1.** Zip folder containing the uncropped gel shown in *Figure 1—figure supplement 1E* with the relevant bands clearly labeled.

**Figure supplement 1—source data 2.** Excel file containing the source data for panels I, K–M.

~48%) (*Figure 1G* and *Figure 1—source data 1*). We further used the momentum scaling spectrum (MSS) analysis to classify the type of motion of the mobile trajectories (*Ferrari et al., 2001*; *Sbalzarini and Koumoutsakos, 2005*). We found a significant increase in the fraction of confined trajectories on mDCs as compared to iDC (~47% vs. ~35%), and no differences in the fraction of free mobile trajectories between both cell types (*Figure 1H* and *Figure 1—source data 1*). Moreover, immobile and/or confined trajectories in mDC were preferentially located close to the cell edges (*Figure 1I, J*) as compared to the more even distribution observed in iDC, suggesting that upon DC activation Siglec-1 diffusion becomes spatially constrained in specific regions of the plasma membrane. Altogether, these results indicate that LPS activation of DCs not only increases Siglec-1 expression, but importantly, it also modulates its nanoclustering, lateral mobility, and overall spatial distribution on the plasma membrane.

## Formin-dependent actin polymerization regulates Siglec-1 nanoscale organization and mobility

Previous studies have demonstrated an essential role for the actin cytoskeleton modulating the capture of HIV-1 and the formation of the VCC by mDCs (*Izquierdo-Useros et al., 2011*; *Ménager and Littman, 2016*). Although Siglec-1 does not have known intracellular motifs of actin interaction (*Bornhöfft et al., 2018*), it has been documented that the actin cytoskeleton can regulate the nanoscale organization of different membrane receptors, and even lipids, that do not directly bind to actin (*Freeman et al., 2018*; *Kusumi et al., 2012*; *Rottner et al., 2017*). To assess the role of actin in the spatial organization of Siglec-1, we performed dual color STED imaging of Siglec-1 together with actin in iDCs and mDCs (*Figure 2A*). When we traced the intensity distributions of actin and Siglec-1 from the cell center we observed a good spatial correlation of both markers in iDC and mDC (*Figure 2B* and *Figure 2—source data 1*). However, consistent with our previous analysis showing the preferential distribution of confined Siglec-1 molecules close to the cell edges, Siglec-1 and actin were highly enriched near the edges of mDC as compared to iDC, suggesting a role for actin in the regulation of Siglec-1 nanoclustering and its spatial confinement.

To further test this hypothesis, we first treated mDCs with Cytochalasin D (CytoD), a drug that inhibits actin polymerization. In these conditions, Siglec-1 staining dispersed throughout the plasma membrane (*Figure 2—figure supplement 1A, B* and *Figure 2—figure supplement 1—source data*

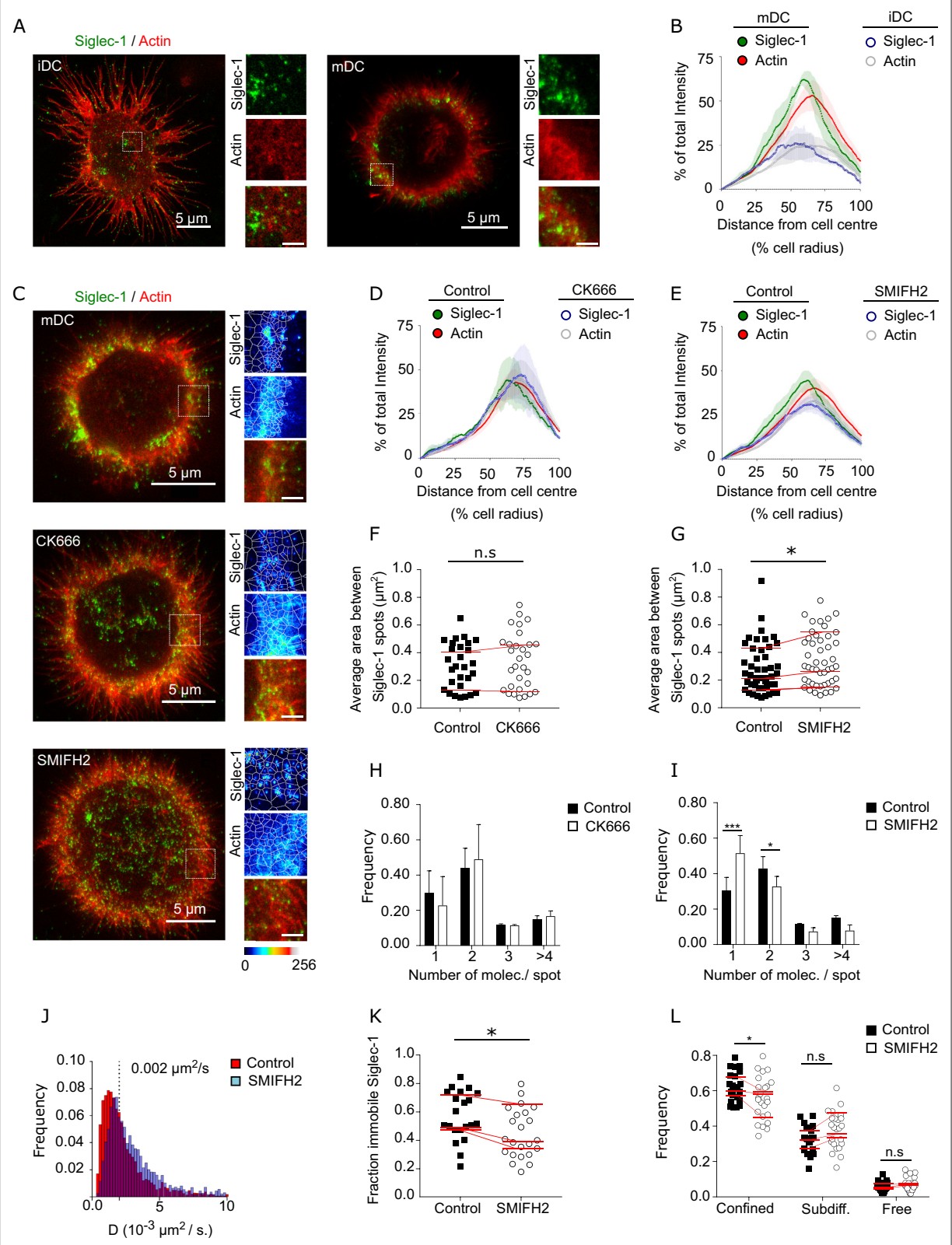

**Figure 2.** Formin-dependent actin polymerization regulates Siglec-1 nanoscale organization and mobility. (**A**) Representative dual color STED images of immature dendritic cell (iDC; left) and mature dendritic cell (mDC; right) stained with Siglec-1 and actin (labeled with SiR-actin). Insets show magnified areas with pseudo-color-coded intensities of Siglec-1 (upper), actin (middle), and merged image (lower) (scale bar: 1 μm). (**B**) Siglec-1 and actin intensity from the cell center toward the edge expressed as percentage of total intensity for each marker in iDC and mDC. Lines represent the mean ± standard

*Figure 2 continued on next page*

*Figure 2 continued*

error of the mean (SEM) of 3 different donors (minimum of 9 cells/donor and condition). (**C**) Representative STED images of mDC stained for Siglec-1 and actin, together with enlarged insets, in control conditions and after 1-hr treatment with CK666 (100 µM) or SMIFH2 (25 µM). The white lines in the insets delimit the Voronoi areas between adjacent Siglec-1 spots; the smaller the areas, the closer single Siglec-1 spots reside from each other (scale bars insets: 1 µm). Siglec-1 and actin intensity from the cell center toward the edge expressed as percentage of total intensity for each marker in control mDC and treated with CK666 (**D**) or SMIFH2 (**E**). Lines represent the mean ± SEM of 2 (**D**) and 4 (**E**) different donors (minimum 10 cells/donor and condition). (**F**) Average Voronoi areas between contiguous Siglec-1 spots per cell in control and CK666-treated cells (2 donors, at least 10 cells/donor). Red lines connect the average of all the cells in each donor. (**G**) Similar to (**F**) for control and SMIFH2-treated cells (3 donors, at least 10 cells/donor). (**H**) Frequency histogram of the number of Siglec-1 molecules per spot in control and CK666-treated mDC. Mean ± SEM of a minimum of 10 cells/donor from 2 donors. (**I**) Similar to (**H**) for control and SMIFH2-treated cells (3 donors, at least 10 cells/donor). (**J**) Frequency of Siglec-1 diffusion coefficients in control mDC (*n* = 8 cells, 4424 trajectories) and cells treated with SMIFH (*n* = 8 cells, 5002 trajectories) from one donor. (**K**) Fraction of immobile trajectories in control and SMIFH2-treated mDCs. A total of 21 (control) and 24 (SMIFH2) cells were quantified from 3 donors (minimum 6 cells and 1261 trajectories/donor and condition). Red lines connect the average of each individual donor. (**L**) Fraction of mobile trajectories classified as confined, sub-diffusive or free in control and SMIFH2-treated mDC. Each symbol corresponds to an individual cell, red lines show the average value of all cells from 3 donors (minimum 6 cells/donor and condition). ns, p > R 0.05, * p < % 0.05, *** p < % 0.0001.

The online version of this article includes the following source data and figure supplement(s) for figure 2:

**Source data 1.** Excel file containing the source data for *Figure 2B, D–L*.

**Figure supplement 1.** Formin-dependent actin polymerization regulates Siglec-1 nanoscale organization and mobility.

**Figure supplement 1—source data 1.** Excel file containing the source data for panels B, C, E, I.

*1*) and the fraction of Siglec-1 receptors found in dimers or small nanoclusters decreased in favor of a significant increase of monomers (*Figure 2—figure supplement 1C* and *Figure 2—figure supplement 1—source data 1*). Because cortical actin mainly depends on two different polymerization mechanisms, one Arp2/3 dependent which forms branched actin, and another dependent on formins which produce filamentous actin bundles (*Rottner et al., 2017*), we selectively inhibited each of these mechanisms using CK666 (Arp2/3 inhibitor) or SMIFH2 (formin inhibitor) in mDCs (*Figure 2C*). The association of Siglec-1 high-density regions with cortical actin near the cell edges was not affected by Arp2/3 inhibition, but decreased when we inhibited formins (*Figure 2D, E* and *Figure 2—source data 1*). Such dispersion was further quantified using a tessellation-based analysis to define areas between adjacent Siglec-1 spots (the smaller the areas, the higher the density). Whereas Arp2/3 inhibition did not affect the proximity between Siglec-1 spots (*Figure 2F* and *Figure 2—source data 1*), formin inhibition significantly increased the separation between individual spots consistent with their spatial dispersion (*Figure 2G* and *Figure 2—source data 1*). Importantly, formin inhibition, but not Arp2/3 inhibition, reduced the basal Siglec-1 nanoclustering as compared to control conditions (*Figure 2H, I* and *Figure 2—source data 1*).

These results indicate that Siglec-1 nanoclustering in activated DCs relies on a formin-dependent actin polymerization mechanism of confinement. These findings were further substantiated by SPT analysis. Although the mobile fraction of Siglec-1 (above 0.002 µm$^2$/s) did not show significant differences in the diffusion coefficients (*Figure 2J* and *Figure 2—source data 1*), we observed a significant decrease in the fraction of immobile and confined trajectories upon formin inhibition (*Figure 2K, L* and *Figure 2—source data 1*). Moreover, consistent with the dissociation of Siglec-1 from the regions enriched in cortical actin, formin inhibition also led to a more even distribution of immobile and mobile Siglec-1 receptors through the cell area (*Figure 2—figure supplement 1D, E*).

To further demonstrate the relationship between formins and Siglec-1 nanoscale organization, we measured Siglec-1 distribution in a physiological situation in which different types of actin pools simultaneously occur at specific regions of the plasma membrane. Activated DCs in confined environments, when subjected to chemokine stimuli, migrate in a process that requires actin polarization at the leading edge (mainly dependent on Arp2/3 branched actin) and at the cell rear (formin-dependent filamentous actin) (*Jolly et al., 2007*; *Lämmermann et al., 2009a*; *Paluch et al., 2016*; *Vargas et al., 2016*). We thus squeezed activated DCs between a layer of agarose and a coverslip with a homogeneous concentration of the chemokine CCL19, and subsequently fixed and stained the cells for Siglec-1 and actin. The spatial constrain of mDCs in this semi-3D environment was optimal to observe in the same plane a clear polarization of a protrusive lamellipodium and a contractile trailing edge (*Figure 2—figure supplement 1F*). Whereas actin staining was isotropic at the front and cell rear (*Figure 2—figure supplement 1G*), Siglec-1 was remarkably

enriched at the formin-dependent trailing edge (*Figure 2—figure supplement 1H*). Moreover, a significant enhancement of Siglec-1 nanoclusters was detected at the trailing edge as compared to the lamellipodium at the front (*Figure 2—figure supplement 1I* and *Figure 2—figure supplement 1—source data 1*). Collectively, these results indicate that formin-dependent cortical actin regulates the nanoscale organization, membrane lateral confinement and diffusion of Siglec-1 in activated DCs.

## Siglec-1 confinement and nanoclustering occurs in polarized regions of the plasma membrane characterized by RhoA activity

The previous results suggest differences in the organization of the actin cytoskeleton between mDCs and iDCs, which might impact on the overall spatiotemporal organization of Siglec-1. Recent studies have shown that DC maturation increases Rho activity in polarized regions of the membrane, which in turn enhances formin-dependent actin polymerization in filamentous bundles (*Vargas et al., 2016*). Thus, to assess the existence of a basal polarization of different actin polymerization mechanisms regulating the confinement of Siglec-1, we first imaged the distribution of Siglec-1 together with actin and the phosphorylated form of ezrin–radixin–moesin (pERM), a well-known actin-associated marker that becomes activated downstream of RhoA (*Figure 3A*). As previously reported in LPS-activated macrophages (*Di Pietro et al., 2017*), we detected an increase in the levels of pERM in mDCs as compared to iDCs (*Figure 3—figure supplement 1A* and *Figure 3—figure supplement 1—source data 1*). Most importantly, we observed a clear polarization of Siglec-1 to cell regions closer to the basal membrane in mDCs (i.e., within the first 3–5 µm) (*Figure 3B* and *Figure 3—source data 1*), and characterized by the abundance of filopodia enriched in pERM (*Figure 3C, D* and *Figure 3—source data 1*).

Next, we assessed if Rho activity was involved in the polarized distribution of Siglec-1 in mDCs. Pharmacological inhibition of RhoA (CT04) as well as of its downstream effector, the Rho-associated protein kinase ROCK (Y-27632), decreased the levels of pERM in mDCs (*Figure 3—figure supplement 1B, C* and *Figure 3—figure supplement 1—source data 1*) and importantly, disrupted the polarized confinement of Siglec-1 in mDCs, resulting in a more even distribution throughout the plasma membrane (*Figure 3E–G* and *Figure 3—source data 1*). Moreover, RhoA and ROCK inhibition led to a significant decrease in the density of Siglec-1 nanoclusters, as determined by STED (*Figure 3E*, bottom panels; *Figure 3H, I* and *Figure 3—source data 1*), and to a reduced basal nanoclustering (*Figure 3J, K* and *Figure 3—source data 1*), with values similar to those measured in iDCs. Significantly, treatment of iDCs with an RhoA activator (CN03), led to an increase of pERM levels (*Figure 3—figure supplement 1D, E* and *Figure 3—figure supplement 1—source data 1*), Siglec-1 and pERM polarization in the basal plane of the cells (*Figure 3—figure supplement 1F, G* and *Figure 3—figure supplement 1—source data 1*) and induction of basal Siglec-1 nanoclustering (*Figure 3—figure supplement 1H* and *Figure 3—figure supplement 1—source data 1*) fully mirroring the results obtained on activated DCs. Together, these data strongly indicate that basal nanoclustering of Siglec-1 on activated DCs is the consequence of higher basal levels of RhoA activation, as compared to resting DCs.

It has been reported that the spatiotemporal organization of several receptors is influenced by the presence of additional membrane receptors that work as pickets in association with pERM (*Freeman et al., 2018*; *Kalay et al., 2014*; *Kusumi et al., 2012*; *Trimble and Grinstein, 2015*). Given our results, we thus checked the effects of ezrin inhibition in the spatial organization of Siglec-1 on mDCs. Pharmacological inhibition of ERM (NSC668394) led to a significant drop in the levels of pERM (*Figure 3—figure supplement 1I, J* and *Figure 3—figure supplement 1—source data 1*), significantly disrupted the spatial polarization of Siglec-1 (*Figure 3—figure supplement 1K* and *Figure 3—figure supplement 1—source data 1*) and reduced Siglec-1 nanoclustering (*Figure 3—figure supplement 1L* and *Figure 3—figure supplement 1—source data 1*). Notably, formin inhibition did not alter the basal levels of pERM (*Figure 3—figure supplement 1M* and *Figure 3—figure supplement 1—source data 1*) but disrupted the polarized distribution of Siglec-1 (*Figure 3—figure supplement 1N* and *Figure 3—figure supplement 1—source data 1*). Hence, our results strongly indicate that Rho activity in mDCs regulates Siglec-1 basal nanoclustering and its spatial confinement in polarized regions of the plasma membrane through two necessary, but independent mechanisms: formin activation and ROCK-mediated ERM phosphorylation (*Figure 3L*).

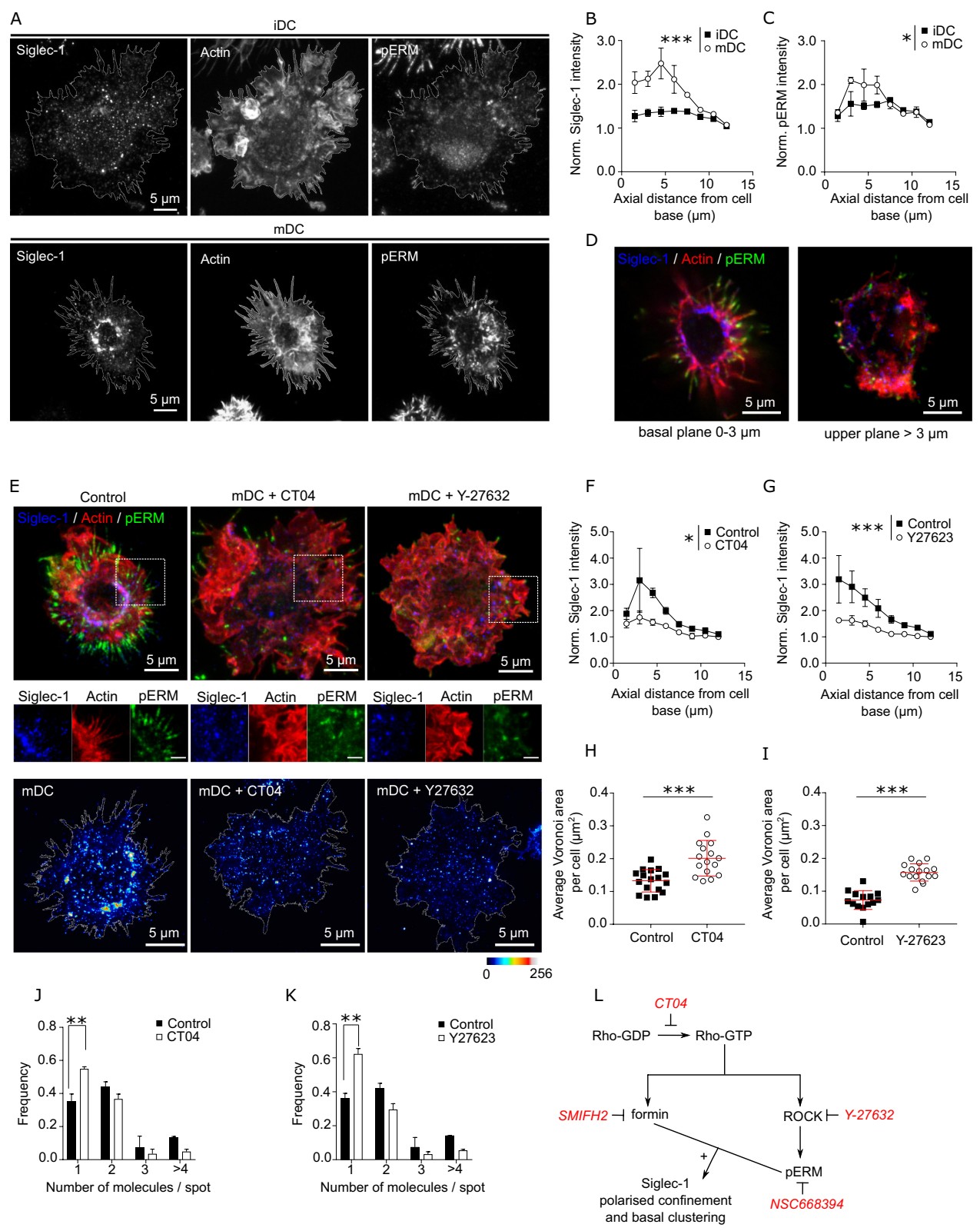

**Figure 3.** Siglec-1 confinement and basal nanoclustering occur in polarized regions of the plasma membrane characterized by RhoA activity. (**A**) Representative maximum intensity projections of 3D confocal images of immature dendritic cell (iDC; upper row) and mature dendritic cell (mDC; lower row) immunostained with anti-Siglec-1, SiR-actin, and pERM. Plots of the axial distribution from the cell base of Siglec-1 (**B**) and pERM (**C**) intensities. Values in each plane show the mean intensity in steps of 1.5 μm from the cell base, normalized by the minimum mean intensity within the whole stack.

*Figure 3 continued on next page*

**Figure 3 continued**

Each symbol represents the mean ± standard error of the mean (SEM) from two donors (at least 10 cells per condition). (**D**) Confocal projections of the basal (0–5 μm) and the apical (>5 μm) parts of the mDC example shown in (**A**). (**E**) Top panels: confocal maximum intensity projections images of control mDC and cells treated for 3 hr with CT04 (2 μg/ml), or for 1 hr with Y-27632 (30 μM), fixed and immunostained with anti-Siglec-1, SiR-actin, and pERM. Middle panels: enlarged images from the highlighted regions of the top images (scale bar: 2 μm). Bottom panels: STED images of mDCs treated as described above and immunostained with anti-Siglec-1. Siglec-1 intensity as function of axial distance from the cell base in control mDC and cells treated with CT04 (**F**) and Y-27623 (**G**). Results show the mean ± standard error of the mean (SEM) of two donors, with at least 7 (**F**) and 9 (**G**) cells/ donor and condition. Average Voronoi areas between contiguous Siglec-1 spots in control mDC, and cells treated with CT04 (**H**) and Y-27623 (**I**). Each dot shows the average Voronoi area per cell; results show the average ± SD of one representative experiment (n = 2) with a minimum of 16 (**H**) and 14 (**I**) cells. Number of Siglec-1 molecules per spot in control mDCs and cells treated with CT04 (**J**) and Y-27623 (**K**). Mean ± SEM of 2 donors, with a minimum of 10 (**J**) and 9 (**K**) cells/donor. (**L**) Scheme of the different pathways downstream of Rho activation targeted by different inhibitors. Statistics in the legends of panels B, C, F, G correspond to the significance of a two-way analysis of variance (ANOVA) test depending on maturation status (iDC vs. mDC in **A**, **B**) or treatment control vs. CT04 or Y-27623 in **F**, **G**. * p < % 0.05, ** p < % 0.001; *** p < % 0.0001.

The online version of this article includes the following source data and figure supplement(s) for figure 3:

**Source data 1.** Excel file containing the source data for *Figure 3B, C, F–K*.

**Figure supplement 1.** Siglec-1 confinement and basal nanoclustering occurs in polarized regions of the plasma membrane characterized by RhoA activity.

**Figure supplement 1—source data 1.** Excel file containing the source data for panels B, C, E–H, J–N.

## Siglec-1 basal nanoclustering increases the capture of HIV-1 viral-like particles and the avidity to gangliosides carrying sialic acid ligands

Previous reports have highlighted the importance of nanoscale organization of viral receptors on the capture of viral particles by providing multimolecular docking platforms with high avidity for viral ligands (*Cambi et al., 2004*). To assess the relevance of Siglec-1 nanoscale organization in the capture of HIV-1, we pulsed control, SMIFH2-, or CT04-treated mDCs for 5 min with viral-like particles (VLPs). Inhibition of formin-mediated actin polymerization (*Figure 4A, B* and *Figure 4—source data 1*) or Rho activity (*Figure 4—figure supplement 1A, B* and *Figure 4—figure supplement 1—source data 1*), respectively, which disperse Siglec-1 nanoclustering and its spatial distribution, significantly decreased the immediate binding of VLPs to Siglec-1. Confocal stacks taken from cells immunostained with Siglec-1 without permeabilization revealed no significant differences in the total plasma membrane levels of Siglec-1 in control, SMIFH2-, or CT04-treated cells (*Figure 4C*, *Figure 4—source data 1*, *Figure 4—figure supplement 1C*, and *Figure 4—figure supplement 1—source data 1*), indicating that receptor availability was not responsible for the decrease in VLP capture. Remarkably, when we analyzed the nanoscale organization of Siglec-1 on 5 min-VLP-pulsed cells, we found that Siglec-1 nanoclusters colocalizing with VLPs had significantly higher number of molecules per spot as compared to non-colocalizing Siglec-1 nanoclusters, in both control and SMIFH2-treated cells (*Figure 4D* and *Figure 4—source data 1*). These results suggest that both, formin and Rho-dependent Siglec-1 nanoclustering, facilitate the capture of VLPs by providing high-avidity docking platforms to viral ligands.

To further substantiate this hypothesis, we generated artificial Large Unilamellar Vesicles (LUVs) (~150 nm in diameter) mimicking the lipid composition of the HIV-1 envelope, but with a gradient of concentrations of the Siglec-1 sialic acid containing ligand GM-1 (from 0.25% to 4%). We pulsed mDCs for 5 min with LUVs and recorded dual color STED images of Siglec-1 and LUVs labeled with texasRed (tRed) (*Figure 4E*). As expected, the capture of LUVs was strongly dependent on the GM-1 concentration, being highest at 4% GM-1 (*Figure 4F* and *Figure 4—source data 1*). Remarkably, LUVs with low levels of GM-1 (0.25% and 0.5%) showed a significant preference for binding to larger Siglec-1 nanoclusters and with higher number of molecules per spot, as compared to the LUVs with high GM-1 levels (4%) (*Figure 4G, H* and *Figure 4—source data 1*). These data indicate that Siglec-1 clustering contributes to overcome the decrease in ligand availability. Moreover, LUVs preferentially bound to regions with high density of Siglec-1 receptors, an effect that was more pronounced for LUVs carrying low GM-1 levels (*Figure 4I, J* and *Figure 4—source data 1*). As a whole, these results indicate that basal Siglec-1 nanoclustering and aggregation in highly dense regions of the membrane constitutes a physical mechanism to increase HIV-1 capture efficiency by increasing the avidity of Siglec-1 receptors to gangliosides carrying sialic acid ligands.

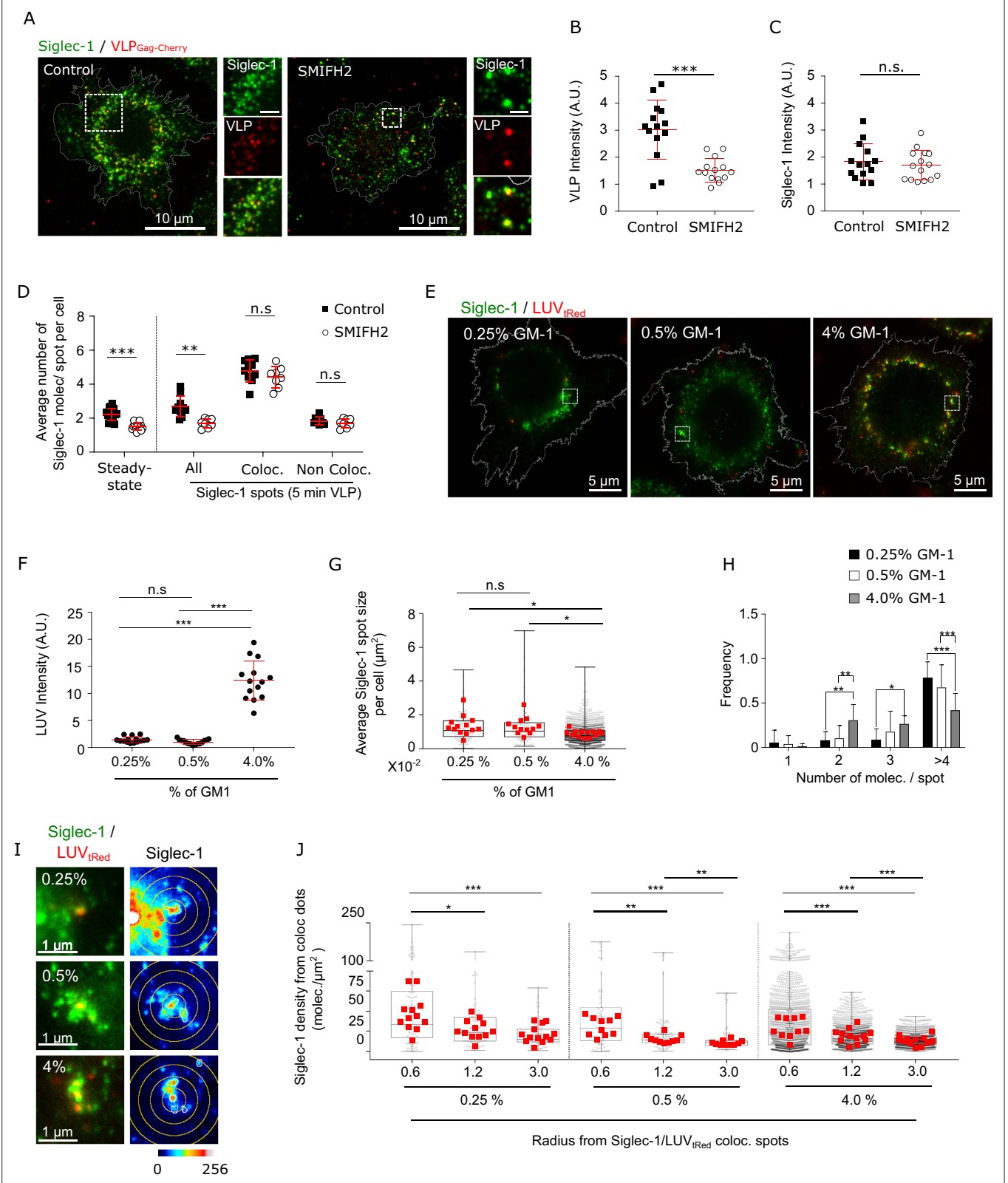

**Figure 4.** Siglec-1 basal nanoclustering increases the capture of HIV-1-VLPs and the avidity to gangliosides carrying sialic acid ligands. (**A**) Confocal images of control mature dendritic cell (mDC), and cells treated with SMIFH2, pulsed for 5 min with VLP-Gag-Cherry and immunostained with anti-Siglec-1. Insets on the right correspond to enlarged views of the regions highlighted in the main images (scale bars: 2 μm). (**B**) VLP Intensity in control and SMIFH2-treated cells, after 5 min of VLP capture. Mean ± standard deviation (SD) of one representative experiment (*n* = 2) with 14 cells per

*Figure 4 continued on next page*

*Figure 4 continued*

condition. (**C**) Levels of Siglec-1 in control and SMIFH2-treated cells, after 5 min of VLP capture. Mean ± SD of one representative experiment (*n* = 2), 14 cells per condition. (**D**) Average number of Siglec-1 molecules per spot per cell in control and SMIFH2-treated mDC, at steady state and after 5 min of VLP capture. Siglec-1 data after 5 min VLP exposure are further separated according to whether Siglec-1 spots colocalize (or not) with VLP particles. Data correspond to the mean + SD of one representative experiment (*n* = 3 for the absence of VLP and *n* = 2 for the 5 min VLP) in which a minimum of 8 cells per experiment and condition were analyzed. (**E**) Representative dual color STED images of mDC immunostained with anti-Siglec-1 after 5-min pulse with tRed-LUVs carrying different GM-1 concentrations. Squares are shown as magnified regions in panel (**I**). (**F**) LUV intensity in mDC after 5 min of LUV capture. Mean ± SD of a minimum of 14 cells per condition. (**G**) Sizes of Siglec-1 spots colocalizing with LUVs. Small dots represent each spot analyzed and red squares correspond to the average size of all colocalizing spots in a cell (at least 12 cells per condition). (**H**) Number of Siglec-1 molecules per spot colocalizing with LUVs. Results correspond to the mean ± SD of a minimum of 12 cells per condition. (**I**) Magnified STED images of Siglec-1 colocalizing with LUVs carrying different GM-1 concentrations. Circles with different radii on the Siglec-1 images are drawn starting from the center of each colocalizing spot. (**J**) Siglec-1 density (i.e., number of Siglec-1 molecules per µm²) at different radius from Siglec-1 spots colocalizing with 0.25–4% GM-1 LUVs (each small dot is an individual colocalizing spot, red squares are the mean values in each cell analyzed, 12 cells per condition). ns, p > R 0.05, * p < % 0.05, ** p < % 0.001; *** p < % 0.0001.

The online version of this article includes the following source data and figure supplement(s) for figure 4:

**Source data 1.** Excel file containing the source data for *Figure 4B–D, F–H, J*.

**Figure supplement 1.** Siglec-1 basal nanoclustering increases the capture of HIV-1-VLPs and the avidity to gangliosides carrying sialic acid ligands.

**Figure supplement 1—source data 1.** Excel file containing the source data for panels B, C.

## Interaction with HIV-1-VLPs induces Siglec-1 changes at the nano- and meso-scale toward the final formation of the VCC

It is known that initial virus binding in mDCs is followed by a progressive polarized movement of the viruses toward the VCC (*Hyun et al., 2008*; *Izquierdo-Useros et al., 2011*). To enquire if iDC and mDC show differences in the events that occur after initial binding of the virus to Siglec-1, we visualized and compared the spatial evolution of Siglec-1 in iDCs and mDCs at different incubation times with VLPs. After 5 min of VLP capture, only a very modest colocalization of VLPs with Siglec-1 was observed in both iDCs and mDCs being slightly higher in the case of mDCs, (*Figure 5—figure supplement 1A* and *Figure 5—figure supplement 1—source data 1*). Colocalization progressively increased in time for both types of DCs, but became significantly higher for mDCs at 60 min of VLP incubation (*Figure 5—figure supplement 1A* and *Figure 5—figure supplement 1—source data 1*). This indicates that Siglec-1 is involved in the initial capture of VLPs by DCs regardless of their activation state. Notably, in contrast to iDCs which showed no significant changes in Siglec-1 clustering during the first 30 min of VLP capture (*Figure 5A , B* and *Figure 5—source data 1*), mDCs already exhibited increased Siglec-1 clustering at these times (*Figure 5C, D* and *Figure 5—source data 1*). The latter was also accompanied by an increase in the sizes of Siglec-1 spots colocalizing with VLPs (*Figure 5—figure supplement 1B* and *Figure 5—figure supplement 1—source data 1*). Moreover, cumulative intensities plots of individual VLPs as a function of distance from their center of mass, showed no changes on iDCs for different capture times (*Figure 5E* and *Figure 5—source data 1*), whereas a significant increase in the proximity, that is, polarization, of VLPs in mDC was observed at 30 min (*Figure 5F* and *Figure 5—source data 1*). These results reveal remarkable changes in the nano- and meso-scale organization of Siglec-1 upon VLP capture on mDCs occurring already at early capture times. In contrast, more modest changes are observed on iDCs and are only detected after 60 min of VLP capture.

The fact that VLP polarization occurs concomitantly to the progressive growth of Siglec-1 clusters colocalizing with VLPs on mDCs suggests the possibility that active Siglec-1 clustering, promoted by their interaction with multiple gangliosides carrying sialic ligands in the HIV-1 membrane, regulates the trafficking of the receptor/virus complexes to the VCC. To investigate this scenario, we pulsed mDCs with LUVs carrying low and high GM-1 concentrations, and assessed the evolution of Siglec-1 nanoclusters and LUV spatial dispersion within the plasma membrane of mDCs along time (representative images shown in *Figure 5—figure supplement 1C*). In the case of high GM-1 concentrations (4%), with more ligand available to interact with Siglec-1 receptors, 30 min of binding led on average to ~1.5-fold increase in the number of colocalizing Siglec-1 molecules per spot with respect to the values measured immediately after binding (5 min) (*Figure 5G* and *Figure 5—source data 1*). In contrast, in the case of LUVs with low (0.5%) GM-1, the number of colocalizing Siglec-1 molecules/spot did not increase to the same level as to the high GM-1 concentration until 60 min (*Figure 5G*

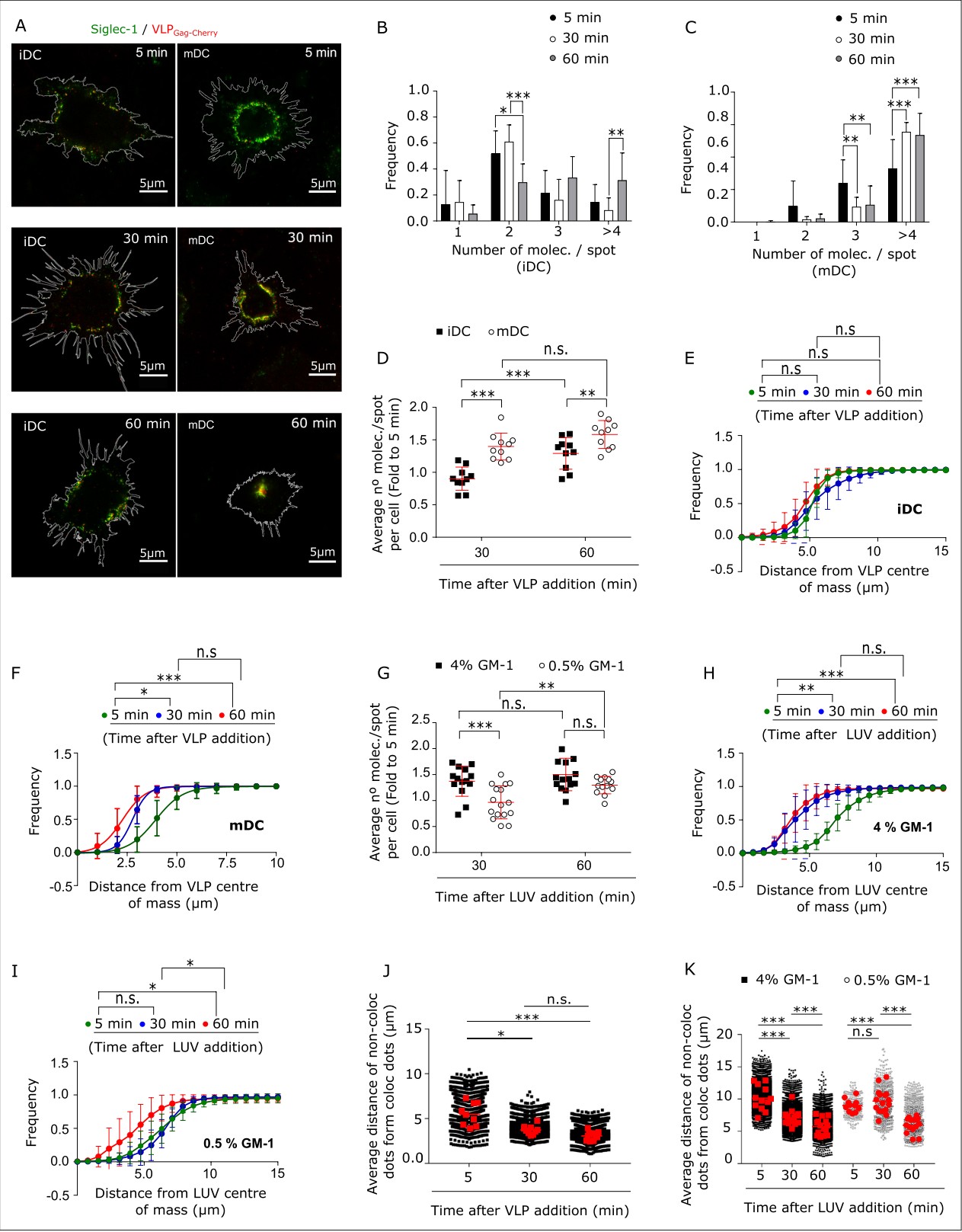

**Figure 5.** Interaction with HIV-1 particles induces Siglec-1 changes at the nano- and meso-scale toward the formation of the virus-containing compartment (VCC). (**A**) Representative STED images of immature dendritic cell (iDC; left) and mature dendritic cell (mDC; right) immunostained with anti-Siglec-1 after different times of VLP-Gag-Cherry capture. Frequency histograms of Siglec-1 molecules/spot colocalizing with VLP-Gag-Cherry in iDC (**B**) and mDC (**C**) at different times of VLP capture. Mean ± standard deviation (SD) of a minimum of 10 cells per incubation time in iDC and mDC.

*Figure 5 continued on next page*

*Figure 5 continued*

(**D**) Fold increase in the number of Siglec-1 molecules per spot with respect to the average value at 5 min. Mean ± SD of a minimum of 10 cells per time condition in iDC and mDC. Cumulative frequency plots of VLP particles colocalizing with Siglec-1 as a function of the distance from the VLP intensity center-of-mass, in iDC (**E**) and mDC (**F**) at different times of VLP capture. Dots show the mean ± SD of a minimum of 9 cells per condition and lines show the sigmoidal fit to the data. (**G**) Same as in D, but in mDC incubated for different times with LUVs at 0.5% and 4% GM-1 concentrations. Mean ± SD of a minimum of 13 cells per condition. (**H–I**) Same as in (**E, F**) but in mDCs incubated for different times with 4% (**H**) and 0.5% (**I**) of GM-1. Dots show the mean ± SD of a minimum of 7 cells per time and condition. (**J**) Box plots showing the distance of non-colocalizing spots to colocalizing Siglec-1/VLP spots in mDC for different incubation times. Small dots show the average distance of the whole population of non-colocalizing spots from each Siglec-1/VLP colocalizing spot, and red squares denote the average of all colocalizing spots in an individual cell (at least 9 cells per time). (**K**) Same as in (**J**) but in mDC incubated with 4% and 0.5% GM-1 LUVs (minimum of 11 cells per time and condition). Statistical analysis in the legends of panels (**E, F, H, I**) corresponds to a one-way analysis of variance (ANOVA) comparing the distance from the center-of-mass of VLPs (**E, F**) or LUVs (**H, I**) at which we recover 50% of the total intensity of VLP-Gag-Cherry or LUVs-tRed. ns, p > R 0.05, * p < % 0.05, ** p < % 0.001; *** p < % 0.0001.

The online version of this article includes the following source data and figure supplement(s) for figure 5:

**Source data 1.** Excel file containing the source data for *Figure 5B–K*.

**Figure supplement 1.** Interaction with HIV-1 particles induces Siglec-1 changes at the nano- and meso-scale towards the formation of the VCC.

**Figure supplement 1—source data 1.** Excel file containing the source data for panels A, B, D.

and *Figure 5—source data 1*). Interestingly, LUVs with high GM-1 concentration exhibited a clear polarization toward the cell center at 30 min of capture (*Figure 5H* and *Figure 5—source data 1*), whereas the low GM-1 LUVs did not show such polarization until 60 min (when the number Siglec-1 molecules per spot colocalizing with LUVs also starts to increase) (*Figure 5I* and *Figure 5—source data 1*). Altogether these results suggest that active Siglec-1 clustering, which is mediated by the sole interaction of Siglec-1 with its ligand GM-1, regulates the trafficking of Siglec-1 receptors toward the later formation of the VCC.

Notably, Siglec-1 receptors not colocalizing with either VLPs or high GM-1 concentration LUVs also showed some degree of polarization toward the cell center after 30 min of incubation, which was reflected by a significant increase in the proximity between non-colocalizing and colocalizing spots along time (*Figure 5J, K* and *Figure 5—source data 1*). In contrast, such effect was only observed at later times (60 min) when cells were pulsed with low GM-1 concentration LUVs (*Figure 5K* and *Figure 5—source data 1*), indicating that at these low concentrations, polarization progressed slower. Finally, we also observed a slight but significant increase in the clustering of non-colocalizing Siglec-1 spots for the cases of VLPs and of high GM-1 concentration LUVs at early capture times, which was only detected at 60 min for low GM-1 LUVs (*Figure 5—figure supplement 1D* and *Figure 5—figure supplement 1—source data 1*). Altogether these results suggest that Siglec-1 enhanced nanoclustering induced by multiple ligand/receptor interactions causes a global nano- and meso-scale rearrangement of the receptor which would then facilitate its trafficking to the VCC.

## Binding of Siglec-1 to HIV-1 particles induces a global reorganization of the actin cytoskeleton facilitating the formation of a single VCC in mDCs

We have shown that Siglec-1 lateral mobility and nanoscale organization at steady state is controlled by the underlying actin cytoskeleton, dependent on formins and Rho activity. To address how the actin cytoskeleton might also regulate the changes in the nano- and meso-scale organization of Siglec-1 upon binding to gangliosides carrying sialic acid ligands, we performed 3D-confocal imaging on fixed and stained mDCs for actin and Siglec-1 after different times of incubation with either VLPs or LUVs carrying high and low GM-1 concentrations (*Figure 6A*). After 30 min of incubation, colocalization of Siglec-1 and VLPs or high GM-1 (4%) LUVs occurred in a ring-shaped compartment within the first 4 µm above the basal membrane (*Figure 6A*, *Figure 6—figure supplement 1A, B* and *Figure 6—figure supplement 1—source data 1*). Moreover, this was accompanied by a clear polarization of the actin cytoskeleton, characterized by a significant reduction of the cell area at the planes where the ring was formed (*Figure 6B* and *Figure 6—source data 1* for VLP, and *Figure 6C–E* and *Figure 6—source data 1* for LUVs) and by the onset of numerous membrane ruffles in the apical parts of the cell (bottom panels in *Figure 6A*). By contrast, pulsing the cells with low GM-1 (0.5%) concentration LUVs, resulted in a significant delay both, in the accumulation of Siglec-1 and LUVs in a narrow ring-shaped compartment (*Figure 6A*), as well as in the constriction of the basal membrane (*Figure 6C–E*

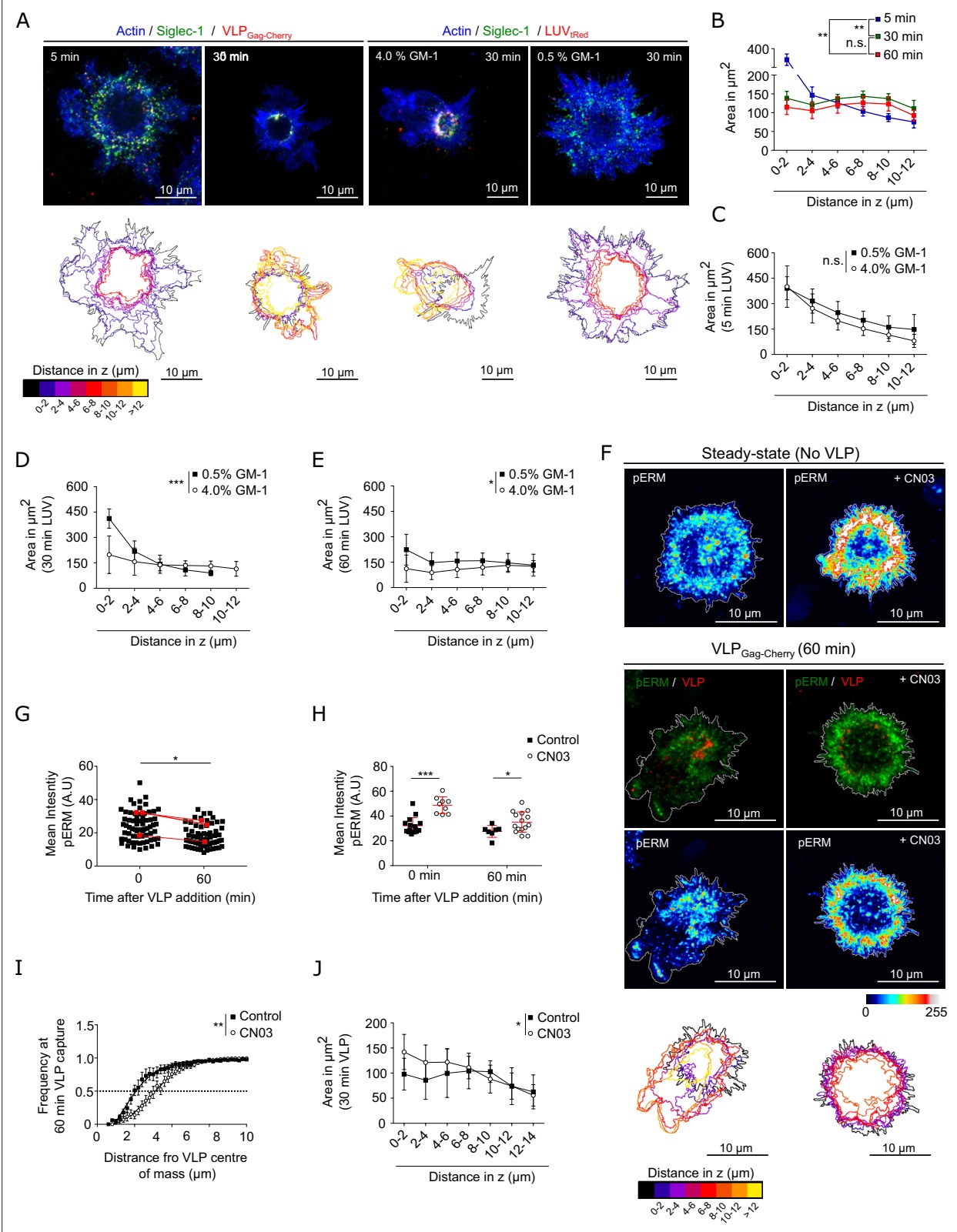

**Figure 6.** Binding of Siglec-1 to HIV-1 induces a global reorganization of the actin cytoskeleton which allows the final formation of a single virus-containing compartment (VCC) in mature dendritic cells (mDCs). (**A**) Confocal maximum intensity projections of 3D confocal images on fixed mDCs immunostained with anti-Siglec-1, SiR-actin and incubated for different times with VLP-Gag-Cherry or tRedLUVs. A ring-shape region or compartment is clearly observed at 30 min of incubation with VLPs or 4% GM1 LUVs, with increased colocalization of Siglec-1, VLPs (or GM1), and actin occurring at axial

*Figure 6 continued on next page*

*Figure 6 continued*

distances of 4–6 µm above the basal membrane plane (see *Figure 6—figure supplement 1A, B* and *Figure 6—figure supplement 1—source data 1for* quantification). Color-coded outlines below each image show the cell perimeter in the axial plane obtained from 3D confocal stacks images (10–15 stacks of 1.5 µm per cell). (**B**) Cell area in the axial plane in mDC for different incubation times with VLPs. Data correspond to the mean ± standard error of the mean (SEM) of 4 (time points 5 and 30 min) and 3 donors (60 min), 7 cells/donor and condition. (**C–E**) Plots as in B for mDC pulsed with LUVs carrying 0.5% and 4% GM-1 for different incubation times (C, 5 min; D, 30 min; E, 60 min). Mean ± standard deviation (SD) of minimum of 7 cells per time and condition. (**F**) Confocal projections of mDC stained with anti-pERM in control conditions (left) or pre-treated for 4 hr with CN03 (right), in the absence (top panels) or the presence of VLP-Gag-Cherry for 60 min (bottom panels). Below scale-colored projections of the cell edges from the bottom to the top of control and CN03-treated mDC after 60 min of VLP capture. (**G**) Average pERM intensity in mDC before and after 60 min of VLP capture (3 donors, at least 7 cells/donor). Red dots are the mean values of each donor. (**H**) Average pERM intensity in control mDC and cells pre-treated with the Rho activator CN03 before and after 60 min of VLP capture. Mean ± SD of a minimum of 7 cells per condition. (**I**) Cumulative frequency plots of VLP intensity as a function of the distance from their center of mass in control mDC and cells pre-treated with CN03 after 60 min of VLP capture. Mean + SD of a minimum of 7 cells per condition. (**J**) Cell area in the axial plane in control and CN03-treated mDC after 60 min of incubation with VLPs. Mean ± SD of a minimum of 7 cells per condition. Statistics in the legends of panels B–E, J correspond to the significance of a two-way analysis of variance (ANOVA) test depending on time after VLP addition (**A**) percentage of GM-1 (**C–E**) or treatment (control mDC vs. CN03-treated cells). Statistics in the legends of panel I correspond to the significance of a Mann–Whitney comparing the distance from the center-of-mass of VLP at which we recover 50% of the total intensity of VLP-Gag-Cherry. ns, $p > R\ 0.05$, * $p < \%\ 0.05$, ** $p < \%\ 0.001$; *** $p < \%\ 0.0001$.

The online version of this article includes the following source data and figure supplement(s) for figure 6:

**Source data 1.** Excel file containing the source data for *Figure 6B–E, G–J*, and *Figure 6G–J*.

**Figure supplement 1.** Binding of Siglec-1 to HIV-1 induces a global reorganization of the acti cytoskeleton which allows the final formation of a single VCC in mDCs.

**Figure supplement 1—source data 1.** Excel file containing the source data for panels A, B, D, E.

**Figure supplement 1—source data 2.** Zip folder containing the original files of the full raw unedited blots corresponding to *Figure 6—figure supplement 1E*.

and *Figure 6—source data 1*). Together, these results show a direct correlation between the ability of VLPs or LUVs (with high GM-1 content) to promote enhanced Siglec-1 nanoclustering and the induction of downstream global actin rearrangements.

We described that Siglec-1 steady-state distribution was mainly found in regions of the plasma membrane regulated by Rho activity and enriched in pERM. Interestingly, when mDCs were pulsed with VLPs we observed a general reduction of pERM as compared to control (i.e., in the absence of VLP) (*Figure 6F, G* and *Figure 6—source data 1*). This drop in pERM suggests a drop in Rho activity and indeed, we observed a similar decay in the levels of the ROCK downstream effectors pMLC (*Figure 6—figure supplement 1C, D* and *Figure 6—figure supplement 1—source data 1*) and p-cofilin (*Figure 6—figure supplement 1E*, *Figure 6—figure supplement 1—source data 1*, *Figure 6—figure supplement 1—source data 2*) after 60 min of VLP capture. Thus, Rho inactivation might be responsible for the major actin reorganization observed and the polarization of VLPs toward the final formation of the VCC.

To test this hypothesis, we pre-treated mDCs with the Rho activator CN03. Activation of Rho led to a homogeneous increase in the levels of pERM as compared to control cells, and such difference remained after 60 min of incubation with VLPs (*Figure 6F, H* and *Figure 6—source data 1*). Importantly, pre-treatment with CN03 delayed the polarization of VLPs into a ring-shape compartment (*Figure 6I* and *Figure 6—source data 1*) and delayed the constriction of the membrane within the planes of VLP accumulation (*Figure 6J* and *Figure 6—source data 1*). Altogether these results strongly indicate that actin remodeling associated to Siglec-1 clustering caused by multiple ligand interactions and the subsequent effect on Rho inactivation is essential for the formation at later stages of the VCC in mDCs.

## Discussion

Several studies have reported the importance of the actin cytoskeleton in almost all the steps of HIV-1 infection (*Audoly et al., 2005*; *Felts et al., 2010*; *Gladnikoff et al., 2009*; *Harmon et al., 2010*; *Iyengar et al., 1998*; *Izquierdo-Useros et al., 2011*; *Jolly et al., 2007*; *Kerviel et al., 2013*; *Li et al., 2017*; *Ménager and Littman, 2016*; *Nikolic et al., 2011*; *Sasaki et al., 1995*; *Shrivastava et al., 2015*; *Wang et al., 2007*). In this study, we demonstrated that RhoA activity and formin-dependent

actin polymerization regulate Siglec-1 organization on the membrane of mDCs by forming receptor nanoclusters that increase the avidity for HIV-1 capture (**Figure 7A**). Our results indicate that differences in actin polymerization mechanisms between iDCs and mDCs are responsible for clustering of Siglec-1 upon DC activation. Along these lines, recent studies have shown that Arp2/3-dependent branched actin is predominant in iDCs, whereas formation of actin bundles by formins is the main mechanism in mDCs (**Vargas et al., 2016**). Accordingly, our results show that in mDCs, Siglec-1 exhibits an uneven distribution, with high-density regions in polarized areas of the plasma membrane that are enriched in the Rho downstream effector pERM, and can be disrupted by the inhibition of RhoA, ROCK, and formins. Interestingly, the abrogation of basal nanoclustering decreases the binding of HIV-1 particles to Siglec-1. Moreover, liposomes with low concentrations of the Siglec-1 ligand GM1 can only bind to large Siglec-1 clusters. These results indicate that the polarized distribution and nanoclustering of Siglec-1 in high-density regions of the membrane may provide very efficient docking platforms capable to interact with HIV-1 and other enveloped viruses even with limited concentrations of gangliosides on their membrane (**Figure 7A**). It is tempting to speculate that mDCs subjected to chemokine gradients in 3D environments, which enhance the polarization of Rho and formin actin polymerization at the trailing edge of migrating cells (**Lämmermann et al., 2009a**; **Lämmermann and Sixt, 2009b**; **Nitschké et al., 2012**; **Vargas et al., 2017**), could increase HIV-1 capture by enhancing the confinement of Siglec-1 receptors at the back of the cell (the uropod). During the natural course of HIV-1 infection there is an increase in the levels of LPS in the serum, mainly due to bacterial translocation induced by damage in the gut epithelium (**Brenchley et al., 2006**). It would be interesting to address if the expected changes in the nanoscale organization of activated DCs during migration from peripheral tissues are maintained once DCs reach the lymph nodes, increasing the efficiency of viral capture and *trans*-infection.

Previous reports have shown that the organization in membrane microdomains of receptors involved in viral binding, like DCSIGN, enhances the avidity of those receptors for their ligands (**Cambi et al., 2004**). However, this effect is largely dependent on the size of the cargo. Siglec-1 is involved in the capture of other cargoes carrying sialylated gangliosides aside from HIV-1 (**Perez-Zsolt et al., 2019**; **Saunderson et al., 2014**). Some of them have sizes considerably larger than HIV-1, like the Ebola virus (>1 μm length), whereas others, like exosomes, exosome-like vesicles, ectosomes, or apoptotic vesicles can have a broad range of sizes below 100 nm (**Théry et al., 2009**). Thus, Siglec-1 nanoclustering may have a different impact in the capture of particulate antigens depending on their structural properties. This possibility can extend the relevance of Siglec-1 spatial organization to other immunological processes like antigen presentation of MHC-I and MHC-II/antigen complexes captured from secreted exosomes (**André et al., 2004**; **Benet et al., 2021**; **Chaput et al., 2004**; **Segura et al., 2007**; **Théry et al., 2002**; **Wakim and Bevan, 2011**).

Our study also underscores the mechanism by which formin-dependent actin polymerization and Rho-ROCK signaling regulates the nanoscale organization of Siglec-1. Our observations on the diffusion behavior of Siglec-1 in mDCs, with a high fraction of molecules displaying immobile or confined motions, suggest that Siglec-1 nanoclustering results from its spatiotemporal confinement in the membrane, which could facilitate the lateral interaction with proximal Siglec-1 receptors. However, Siglec-1 receptors do not possess known cytosolic motifs that directly or indirectly interact with the actin cytoskeleton (**Bornhöfft et al., 2018**). Nevertheless, it is known that the lateral diffusion of lipids and receptors in the plasma membrane can be constrained by interactions with immobile transmembrane proteins or 'pickets', directly connected to the cortical actin through linker proteins like ezrin (**Freeman et al., 2018**; **Kalay et al., 2014**; **Kusumi et al., 2012**; **Trimble and Grinstein, 2015**). The fact that pharmacological inhibition of ROCK and its downstream effector ERM can partially disrupt Siglec-1 nanoclustering and its polarization in highly dense areas of the plasma membrane supports the existence of such protein pickets acting on Siglec-1 (**Figure 7A**). Proteins like CD44 can exert this picket function and confine the diffusion of other transmembrane receptors at zones enriched in pERM and formin-dependent actin polymerization at the trailing edge of mDCs (**Freeman et al., 2018**; **Sil et al., 2019**). It remains to be investigated whether amongst the cofactors needed for the formation of the VCC (**Akiyama et al., 2015**), there are also proteins acting as diffusion barriers to confine Siglec-1 in restricted areas of the cell.

Lastly, we showed that active Siglec-1 clustering promoted by interactions with HIV-1 is essential to regulate its trafficking toward the final formation of the VCC. Importantly, the proximity between

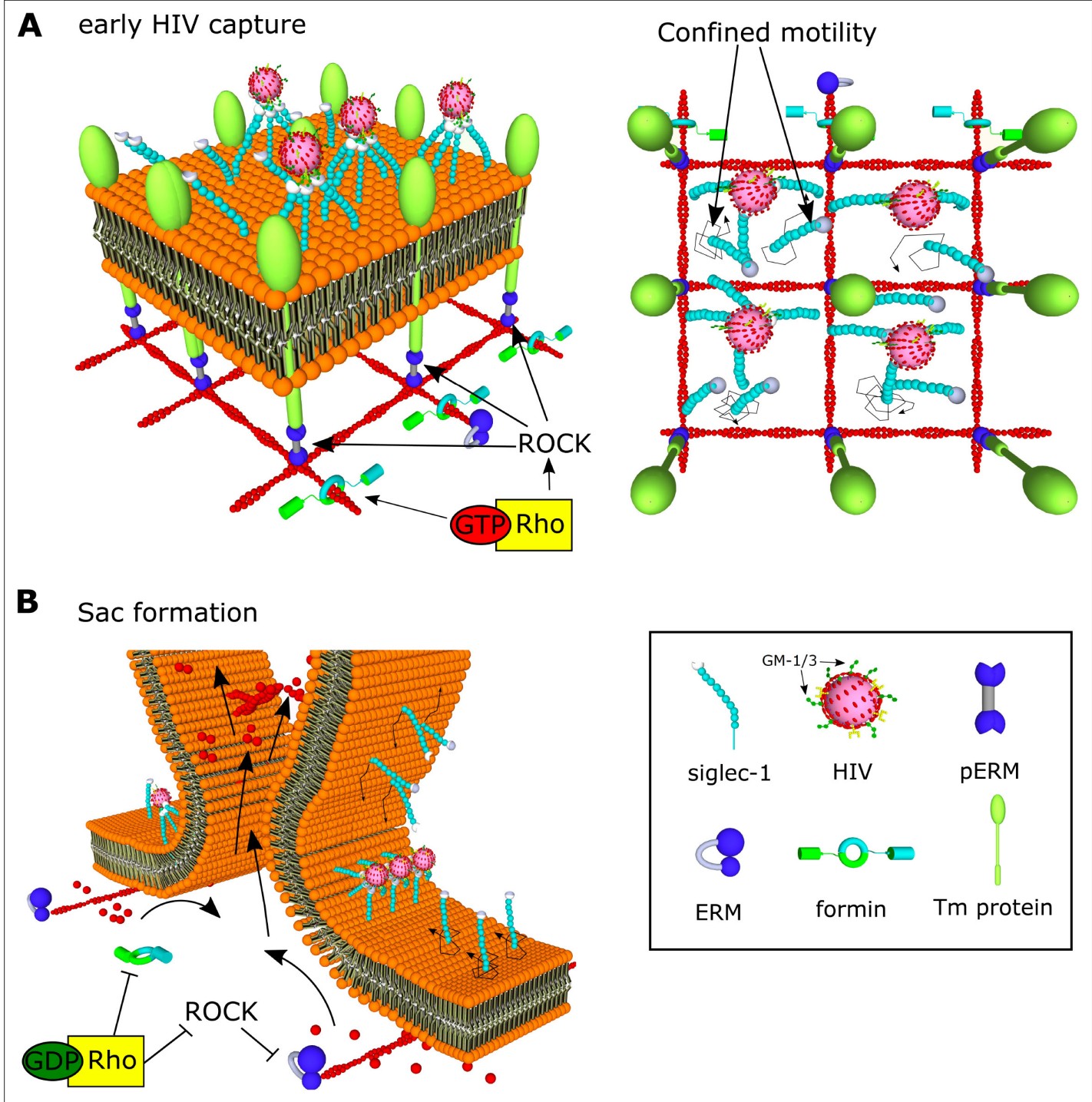

**Figure 7.** Model of HIV-1 capture by Siglec-1 and polarization toward the formation of the virus-containing compartment (VCC) in mature dendritic cell (mDC). (**A**) In steady state, Siglec-1 receptors in mDCs are organized in small nanoclusters proximal to each other. This basal nanoclustering occurs by spatiotemporal confinement of Siglec-1 diffusion into plasma membrane regions dependent on Rho activity, its downstream effector ROCK, pERM, and formin-mediated actin polymerization (see in the middle panel a representation of the cross-section in the 3D reconstruction of a whole cell at early stages of HIV-1 capture). We propose that Siglec-1 mobility is restricted by transmembrane proteins that interact with the filamentous cortical actin cytoskeleton through activated ERM proteins in polarized regions of the mDC membrane (right). Siglec-1 basal nanoclustering is especially relevant to increase the avidity of Siglec-1 for the gangliosides in the membrane of the HIV-1, and therefore regulates the capture capacity of HIV-1 by mDC. (**B**) At late stages of HIV-1 capture, Siglec-1 clustering increases and most receptors accumulate in a polarized ring-shaped compartment that progressively constrains the membrane (see in the middle panel a representation of the cross-section in the 3D reconstruction of a whole cell at late stages of HIV-1 capture). These changes are accompanied by massive actin rearrangements that include extensive membrane ruffling above the plane in which Siglec-1

*Figure 7 continued on next page*

*Figure 7 continued*

and HIV-1 accumulate and by a contraction of the membrane below the ring. This polarization requires a temporal drop in Rho activity. We propose that Rho inactivation would allow for the free diffusion of Siglec-1 through the membrane until being trapped by direct interaction with HIV-1 in the highly dense Siglec-1 regions in which initially most viruses are bound. In parallel, a drop in Rho activity would also lead to actin depolymerization and a decrease in membrane tension together with a retraction of the basal membrane. Both factors would favor the proximity between Siglec-1 bound to HIV-1 clusters and the progressive bending of the membrane until the formation of a single VCC.

VLPs proceeds in parallel with a progressive clustering of Siglec-1, which is much more pronounced in the case of mDCs as compared to iDCs, and dynamically dependent on the concentration of gangliosides in the viral membrane. Indeed, when mDCs are pulsed with liposomes carrying low concentrations of GM-1 we observed a significant delay in the growth of Siglec-1 clusters and a more disperse distribution of these liposomes within the cell membrane, as compared to those with high GM-1 concentrations. In addition, the polarization of viral particles in a ring-shape compartment triggered several morphological changes in mDCs: emergence of membrane ruffles, shrinkage of the basal membrane, and constriction of the cell membrane at the plane where VLPs accumulate. These results indicate that active clustering of Siglec-1 is capable of triggering a downstream signaling cascade that dramatically affects the actin cytoskeleton of mDCs. Accordingly, we showed that VLPs accumulation into the ring-shape compartment coincides with a decay in the levels of the ROCK effectors pERM, pMLC, and p-cofilin. Moreover, pharmacological activation of Rho caused a delay in the constriction of the ring-shape compartment, indicating that Rho inhibition is required for the proper polarization of VLPs toward the final formation of the VCC. A decay in Rho activity may, on the one hand, decrease actin polymerization by formins and activate severing of filamentous actin by cofilin (*Sit and Manser, 2011*), which could explain the retraction of the basal membrane. On the other hand, the inhibition of pERM, one of the main regulators of membrane tension (*Liu et al., 2012a*; *Rouven Brückner et al., 2015*) might facilitate the observed constriction of the membrane at the plane where the virus accumulate. Both phenomena would enhance the proximity between Siglec-1/HIV-1 complexes facilitating the lateral fusion of molecules until a VCC is formed (*Figure 7B*). Interestingly, it has been shown that monocyte-derived macrophages (MDMs) express Siglec-1 on the cell surface and that these cells are able not only to capture VLPs but also to rapidly internalize particles into the VCC in a Siglec-1 and ganglioside-dependent manner (*Hammonds et al., 2017*). It is thus tempting to speculate that similar to mDCs as shown here, macrophages could also use an actin-dependent mechanism to increase the avidity of Siglec-1 to VLPs via nanoclustering of the receptor and for orchestrating the earliest events that finally lead to VCC formation.

How Siglec-1 clustering is capable to activate such changes needs further investigation. The fact that the cytosolic domain of Siglec-1 is not essential for the formation of the VCC reinforces the idea that other cofactors might co-cluster with Siglec-1 to activate intracellular signaling cascades. Interestingly, the cross-linking of several tetraspanins, which are highly enriched in the VCC (*Garcia et al., 2005*; *Izquierdo-Useros et al., 2012a*; *Puryear et al., 2013*), can induce actin rearrangements through the modulation of Rho GTPases (*Brazzoli et al., 2008*; *Delaguillaumie et al., 2002*; *Jones et al., 2016*; *Liu et al., 2012b*). Furthermore, the spatial organization of tetraspanins in the so-called tetraspanin-enriched microdomains (TEMs) is also regulated by interactions with certain lipids like cholesterol (*Zimmerman et al., 2016*) and gangliosides (*Odintsova et al., 2006*). Yet, it is not known if some trans-interactions between the tetraspanin web and the gangliosides exposed in the membrane of HIV-1 can also affect the coalescence of multiple TEM (*Delaguillaumie et al., 2004*). The formation of the VCC could be the result of a synergistic cooperation between the capacity of several tetraspanins to modulate positive and negative membrane curvature (*Delaguillaumie et al., 2004*; *Dharan et al., 2023*; *Dharan et al., 2022*; *Odintsova et al., 2006*) and the differential lipid composition of TEMs, which could facilitate domain-induced inward budding to minimize the line energy at the TEM-membrane interface (*Baumgart et al., 2003*; *Lipowsky, 1992*; *Liu et al., 2006*). In this regard, the use of single-particle tracking and super-resolution microscopy can be of great interest to further investigate the molecular composition of Siglec-1/HIV-1 clusters within the course of the VCC formation, as well as to identify new key regulators of this process.

# Materials and methods

## Antibodies and reagents

Mouse monoclonal antibodies to Siglec-1 (Hsn 7D2) and rabbit polyclonal antibodies against cofilin and phospho-cofilin (phospho S3) were obtained from Abcam. Rabbit antibodies against pERM (48G2) and pMLC were purchased from Cell Signaling. Anti-Mouse IgG F(ab) ATTO488 was from Hypermol and Anti-Rabbit IgGF(ab) Cy3 was from Jackson Immunoresearch. Donkey anti-mouse AlexaFluor Plus 488 secondary antibody was from Thermo Fisher. The SIR-Actin Spirochrome Kit (CY-SC001), the Rho Inhibitor I ADP ribosylation of Rho Asn-41 (CT04) and Rho Activator II (CN03) were obtained from Cytoskeleton. The phosphor-ezrin inhibitor NSC668394 was from Calbiochem. The pan-formin inhibitor SMIFH2, the Arp2/3 inhibitor CK-666 and the actin polymerization inhibitor CytoD were purchased from SIGMA. The ROCK inhibitor Y-27632 was obtained from Merk Millipore.

## Primary cell cultures

Peripheral blood mononuclear cells (PBMCs) were obtained with a Ficoll-Hypaque gradient (Alere Technologies AS) from HIV-1-seronegative donors, and monocyte populations (>97% CD14$^+$) were isolated with CD14-positive selection magnetic beads (Miltenyi Biotec). DCs were obtained by culturing these cells in the presence of 1000 IU/ml granulocyte-macrophage colony-stimulating factor and interleukin-4 (both from R&D) for 7 days and replacing media and cytokines every 2 days. Activated DCs were differentiated by culturing iDCs at day 5 for 2 more days in the presence of 100 ng/ml LPS (Sigma-Aldrich) to induce Siglec-1 expression.

## Plasmids and viral like particle (VLP) stock production

HIV-1-Gag-mCherry-env HXB2 were obtained by co-transfection 1:1 (wt/wt) of the molecular clones pcDNA HIV Gag Cherry (kindly provided by Dr. Gummuluru; 15 µg) and pcDNA HIV-1 env HXB2 (15 µg). HEK-293T cells were transfected with calcium phosphate (Clontech) in T75 flasks using a total of 30 µg plasmid DNA. Seventy-two hours post-transfection, supernatants containing virus were cleared of cellular debris by centrifugation, filtered (0.45 µm; Millipore) and frozen at −80°C until use.

## Generation of LUVs

LUVs were prepared following the extrusion method described in *Mayer et al., 1986*. Lipids were purchased from Avanti Polar Lipids and gangliosides were obtained from Carbosynth. The LUV-tRed lipid composition was: POPC 25 mol%: 1,2-dipalmitoyl-sn-glycero-3-phosphocholine (DPPC), 16 mol%: brain sphingomyelin (SM), 14 mol%: cholesterol (Chol), 39 mol %: GM-1 from 4 to 0.25 mol%. 4% of gangliosides content corresponds to approximately 12,000 gangliosides, mimicking the lipid composition of HIV-1 or VLPs (*Brügger et al., 2006*; *Chan et al., 2008*). Reducing the ganglioside content allowed us to test the role for Siglec-1 nanoclustering in increasing the avidity of receptors to gangliosides carrying sialic acid ligands. All the LUVs contained 2 mol% of 1,2-dihexadecanoyl-sn-glycero-3-phosphoethanolamine (DHPE)-tRed (Molecular Probes). Lipids were mixed in chloroform:methanol (2:1) and dried under nitrogen. Traces of organic solvent were removed by vacuum pumping for 1–2 hr. Subsequently, the dried lipid film was dispersed in N-2-hydroxyethylpiperazine-N'-2-ethanesulfonic acid (HEPES)- sodium buffer and subjected to ten freeze-thaw cycles prior to extruding ten times through two stacked polycarbonate membranes with a 100-nm pore size (Nucleopore, Inc) using the Thermo-barrel extruder (Lipex extruder, Northern Lipids, Inc). To perform mDC pulse with equal concentrations of LUV displaying similar fluorescence intensities, tRed containing LUVs concentration was quantified following the phosphate determination method (*Böttcher et al., 1961*) and the fluorescence emission spectra were recorded setting the excitation at 580 nm in a SLM Aminco series 2 spectrofluorimeter (Spectronic Instruments).

## Immunoblot

DCs derived from PBMCs treated with LPS for 48 hr were seeded at a concentration of $0.5 \times 10^6$ cells well in 12-well plates coated with poly-L-lysine (30 µg/ml). After 1 hr of adhesion cells were pulsed with 50 µl of VLP-Gag-Cherry particles in each well, and gently scrapped on ice after different times of incubation at 37°C. Cell pellets were lysed in Triton X-100 lysis buffer (Thermo Fisher) supplemented with protease inhibitors (Roche), and 10 µg of protein samples were prepared adding NuPAGE LDS

Sample Buffer (Thermo Fisher) with 2.5% beta-mercaptoethanol (Sigma). Samples were further denatured at 80°C for 5 min and loaded in 12% acrylamide gels (Bio-Rad). Electrophoresis was carried in Tris-Acetate SDS running buffer (Bio-Rad) at 100 V and transfer to nitrocellulose membranes was done in Towbin buffer (25 mM Tris, 192 mM glycine, 20% (vol/vol) methanol) for 90 min at 100 V. Membranes were incubated with primary and HRP-secondary antibodies in 1X Tris-Buffered Saline, 0.1% Tween 20 Detergent (TBST)buffer with 5% milk (room temperature for 1 hr or overnight at 4°C) and revealed with an ECL solution (BIO-Rad) following the manufacturer's instructions.

## Generation of single-chain antibodies

Monovalent mouse monoclonal [HSn 7D2] anti-human Siglec-1 Abs were prepared by reduction of the full antibody with Dichlorodiphenyltrichloroethane (DTT) at 10 mM for 30 min at room temperature. DTT was removed dialyzing the samples in phosphate-buffered saline (PBS) using Zeba Spin Desalting Columns (Thermo Scientific) and when required monovalent antibodies were further concentrated using Centricon concentrators according to the manufacturer's instructions. To verify the generation of single-chain antibodies we prepared samples of whole and reduced antibodies with NuPAGE LDS Sample Buffer (4×) and run them in NuPAGE 7% Tris-Acetate Protein Gels (Bio-Rad) in denaturing, non-reducing conditions. Electrophoresis was performed in Tris-Acetate SDS running buffer (Bio-Rad) at 100 V and gels were stained with Coomassie for 1 hr at room temperature (see *Figure 1—figure supplement 1E* and *Figure 1—figure supplement 1—source data 1*).

## Sample preparation for STED and confocal imaging in fixed cells

$1.5 \times 10^5$ immature and LPS (100 ng/ml for 48 hr) matured DCs were plated on glass slides coated with poly-lysine (10 µg/ml). After DCs adhesion to culture plates, cells were pulsed with either VLP-Gag-Cherry particles (30 µl of concentrated VLPs per $1.5 \times 10^5$ cells plated in 270 µl RPMI, final VLP concentration ~2.5 ng per $10^5$ cells) or LUVs carrying different concentration of GM-1 (200 µM) for 5 min to 1 hr at 37°C. After extensive washing, cells were fixed in 4% PFA (10 min at room temperature) and rinsed with $NH_4Cl$ (50 mM in PBS). When needed, permeabilization was done with saponin 0.01% in PBS (10 min, RT). Immunostaining of Siglec-1was done incubating primary and secondary antibodies (1 hr, room temperature) in PBS 1% bovine serum albumin. Specifically, full antibody (mouse monoclonal anti-Siglec-1 [Hsn 7D2] obtained from Abcam) as well as single-chain antibody were used at a final concentration of 10 µg/ml. Secondary antibody (donkey anti-mouse AlexaFluor Plus 488) was used at a final concentration of 20 µg/ml. SiR-actin was added at 1 µM together with primary antibodies. After several washes in PBS glass slides were covered with Fluoromount aquose mounting media (Sigma). When indicated cells were incubated with CytoD (2 µg/ml), CK666 (100 µM), SMIFH2 (25 µM), and Y-27623 (30 µM) for 1 hr at 37°C in complete RPMI (10% fetal bovine serum [FBS], L-glutamine) prior fixation. NSC668394 (250 µM) was added for 3 hr at 37°C in complete RPMI. CT04 (2 µg/ml) and CN03 (2 µg/mL) were added for 3–4 hr at 37°C in RPMI without FBS. When cells were incubated with VLP-Gag-Cherry the concentration of the drugs was maintained for all the time until fixation.

Control experiments were performed using single-chain Abs to label Siglec-1 (instead of full Abs) to exclude any potential artifacts associated with our labeling strategy. As additional controls, we also tried different fixation protocols in combination with single-chain Ab labeling, specifically: 4% PFA (10 min at room temperature) and 1% PFA + 0.2% glutaraldehyde (GA) (10 min at room temperature). Samples from three different donors were prepared as described above.

## STED imaging

STED super-resolution images of DCs were acquired with a confocal microscope (Leica TCS SP8, Leica Microsystems) equipped with an oil immersion objective (HCX PL APO CS x100, Leica) with a 1.4 numerical aperture. Samples were excited with a white light laser (WLL) at 488 nm (Alexa488), 587 nm (mCherry), 583 nm (tRed), and 633 nm (SiR) fixing the laser power to optimal conditions (5–20%) for each experiment. Emission spectra were in the range of 500–540 nm (Alexa488), 550–650 nm (mCherry and tRed), and 645–661 nm (SiR). STED laser beams intensities at 592 nm (Alexa488) and 775 nm (mCherry, tRed, and SiR) were set to 50% and 100% of their power, respectively. Images were acquired with a format of 1024 × 1024 pixels at 400 Hz with a pixel size of ~14–30 nm, adjusting frame

accumulation (1–5) and line average (1–5) to a non-saturating signal for each staining, keeping the same conditions for all the cells analyzed in one experiment.

## Quantification of number of molecules per spot

To define the area of Siglec-1 individual spots STED images were processed using Fiji to apply a subtraction of the background and a Gaussian blur filter (sigma radius 1) followed by a difference of Gaussian (smaller/greater sigma 1:3). Then an intensity threshold was used to create binary masks of the individual spots from which we obtain the mean intensity values per each spot within the original images. Thereafter, we used a MATLAB code (**Manzo, 2017**, https://github.com/cmanzo/DECO) to fit the distribution of intensities of individual spots from antibodies on glass to a lognormal function ($f_1$).

$$f_1 \left( I \right) = \frac{1}{\sqrt{2\pi}\sigma I} e^{-\frac{(\ln I - \mu)^2}{2\sigma^2}}$$

This model corresponds to the expected theoretical distribution for the intensity corresponding to the detection of a fluorescent emitter (**Zanacchi et al., 2017**; **Ménager and Littman, 2016**; **Schmidt et al., 1996**). The intensity values obtained from spots on glass were used to define the $\mu$ (mean) and $\sigma$ (standard deviation) of the lognormal distribution, through its fit to a linear combination of $N = 2$ functions. These parameters were used as a single-molecule reference to define the stoichiometry of the fluorescence of Siglec-1 receptors in the cells measured under identical experimental conditions (**Torreno-Pina et al., 2016**; **van Zanten et al., 2009**). As expected, the intensity histograms of Siglec-1 spots on samples showed higher intensities and broader distributions than the antibodies on glass, indicative of a mixture of different populations of nanoclusters composed by a different number of molecules. To calculate the probability distribution of molecules per spot, the intensity histograms of Siglec-1 spots on cells were fitted to a model distribution $g_N(I)$ composed of a linear combination of functions as described in **Martínez-Muñoz et al., 2018**.

$$g_N \left( I \right) = \sum_{n=1}^{N} \alpha_n \cdot f_n \left( I \right),$$

where $f_n$ shows the intensity distribution intensity of a spot containing n receptors, and $\alpha_n$ is the relative weight of this distribution so that $\sum_{n}^{N} \alpha_n = 1$, being $N$ the maximum number of receptors (in our analysis 12) (**Ménager and Littman, 2016**). We considered that the distribution for a spot containing $n$ receptors could be obtained recursively as

$$f_n = f_{n-1} \otimes f_1$$

where ⊗ represents the convolution of the intensity distribution lognormal functions for $n$ = 1, 2, …, 12 (**Ménager and Littman, 2016**; **Schmidt et al., 1996**; **Torreno-Pina et al., 2016**).

## Determination of number of molecules per spot colocalizing with VLPs

To classify Siglec-1 clusters according to their colocalization with VLPs, STED images from each channel were processed as described in quantification of molecules per spot followed by an intensity threshold segmentation of individual spots to create binary masks. The binary images of segmented spots in each channel were overlapped to identify Siglec-1 colocalizing clusters (with an average area overlap for each siglec-1 spot of ~60% ± 27%). Then, we obtained the Siglec-1 mean intensity values of the spots colocalizing with VLPs from the original unprocessed images, which we used to calculate the probability distribution of number of molecules per spot in the Siglec-1/VLP colocalizing fraction as described above (see quantification of the number of molecules per spot).

To quantify the local density of Siglec-1 molecules surrounding the colocalizing spots, we selected all the spots whose centroids were enclosed in different radius from the centroid of each colocalizing spot. We then used their mean intensity values to assign a number of molecules ($n$) to each spot (see quantification of molecules per spot). The density ($d$) of molecules within a radius $r$ was,

$$d_{(r)} = \sum_{n=1}^{N} \frac{z_n \cdot n}{\pi r^2}$$

where $Z_n$ is the total number of spots with $n$ molecules and $N$ the maximum number of molecules per spot.

To quantify the dispersion of VLPs and LUVs colocalizing with Siglec-1 we measured the distance of each individual particles from its center to the center-of-mass of all the Siglec-1 spots. This distance versus the accumulated intensity frequency was fit to a sigmoidal equation from which we obtained the distance covering 50% of the Siglec-1 intensity for each cell, which was used as reference parameter for statistical analysis. To quantify the proximity between Siglec-1 spots colocalizing or not with VLP/LUV, individual Siglec-1 spots were segmented for colocalization as described above and we measured the average distance between the centers of all non-colocalizing spots to the center of each individual colocalizing spot.

## Generation of in silico random images

To generate in silico images with a random distribution we calculated the FWHM of individual spots by averaging the intensity line profiles of multiple spots on glass and fitting the data to a Gaussian function. In this way we generated in-silico spots with a PSF and an amplitude equivalent to the values obtained from the Gaussian fits of the real data. For each cell, the total number of Siglec-1 molecules randomly distributed in a defined cell area was calculated according to the total number of spots ($Z$) quantified in the experimental data and the probability distributions ($\alpha_n$) of $n$ molecules per spot up to a maximum of $N$ = 12 molecules per spot.

$$\sum_{n=1}^{N} \alpha_n * n * Z$$

## Analysis of actin and Siglec-1 distribution

To correlate the intensities of Siglec-1 and cortical actin, we processed STED images for Siglec-1 as described in 'Quantification of molecules per spot'. Then we measured the mean intensity profiles for each channel from the cell center to the edges in concentric radius of 1.3 pixels. Distance from the cell center was normalized as a percentage of the maximum cell radius for each cell, and intensities at each radius were normalized to the sum of all the intensities for all the radius.

## Under-agarose experiments

Under-agarose chemotaxis assays were performed as described elsewhere (*Kopf et al., 2020*; *Lämmermann et al., 2009a*). Briefly, we added ultrapure agarose (Invitrogen) at 4% to RPMI supplemented with 20% FBS and Hank buffered solution (GIBCO) at a 1:2:1 ratio. We poured 1 ml of the mixture on 35-mm glass coverslips previously coated with 30 µg/ml of fibronectin. After polymerization at RT we added $4 \times 10^5$ cells in 30 µl of RPMI between the agarose pad and the glass. We allowed the cells to equilibrate for about 15–30 min and then added on the borders of the agarose pad 1 ml of complete RPMI with 2.5 µg/ml of CCL19. After 90 min at 37°C we directly added to the cells 1 ml of PFA at 8% for 10 min. We gently lifted the agarose pad and processed the sample for immunostaining with SIR-actin and anti-Siglec-1. Two color STED images acquired as previously described were analyzed for the distribution of actin and Siglec-1 staining at the rear and the cell front. Briefly, we defined a 0 angle between the cell center and the center-of-mass of the uropod at the trailing edge, and quantified the percentage of Siglec-1 and actin staining in 40° binning. For the analysis of Siglec-1 molecules per spot, the receptors counted at the rear were all the spots at ±20° (from the 0 angle) and between 140° and 220° for the front.

## Confocal imaging

Confocal images on fixed DCs were obtained with a confocal microscope (Leica TCS SP8, Leica Microsystems) as specified in 'STED images'. Images were acquired with a format of 1024 × 1024 pixels at 400 Hz with the pinhole set at 1.0 A.U. When performing 3D confocal imaging, the whole surface of cells was covered in stacks of 10–15 images from the basal to the apical membrane. We applied intensity thresholds for each staining and calculated the mean intensity values in axial planes of 1.5 µm. For each cell, the minimum mean intensity value within the stack was used to normalize the axial distribution of each staining. To measure the area of cells at different axial planes after VLP/LUV

capture, we delimited the edges of cells stained with SiR-actin applying a binary mask for each plane and quantified the area enclosed in such perimeter.

## Sample preparation and analysis of Siglec-1 single-particle tracking in living cells

$1 \times 10^6$ immature and LPS (100 ng/ml for 48 hr) matured DCs in suspension were incubated in complete RPMI supplemented with 10% FBS and 1 µg/ml of single-chain Siglec-1 antibodies on ice for 30 min, after which cells were washed three times in PBS and resuspended in 500 µl of complete RPMI. Next, cells were seeded on 35-mm glass coverslips (Corning) coated with poly-L-lysine at 30 µg/mL and after 30–60 min of adhesion at 37°C cells we added Fab-Atto488 anti-mouse antibodies at 1 µg/ml for 5 min. Cells were washed several times in PBS and covered with 1 ml of complete RPMI supplemented with 25 mM of HEPES.

Live imaging was done with a confocal microscope (see STED imaging) connected to a temperature control device at 37°C. For the detection of Fab-Atto488, samples were excited with a WLL laser at 488 nm (20% power) setting the emission range at 509–595 nm. Scan laser speed was set at 1000 Hz to record 3 frames per second in a format of 512 × 512 pixels.

The particle tracking analysis was done using the MOSAIC tool of Fiji (*Sbalzarini and Koumoutsakos, 2005*). The implementation of the plugin requires the approximate particle radius $w$ (in our case 3 pixels), the intensity percentile $r$, which is used to select among the local maxima pixels in a radius $w$ the ones that have an intensity above the selected $r$ value (in our case the upper 0.5 percentile), the cut-off intensity score $Ts$ to remove those pixel values below a intensity threshold (in our case set to 0), the maximum displacement in pixels between contiguous frames (in our settings 5 pixels), and the number of future frames that can be considered to re-connect particles (in our case 2, which means that a single particle can disappear for just one frame to be re-connected with another particle in the next frame).

Individual trajectories of a minimum of 20 points were analyzed calculating their moments of displacement $\mu_v$, which for a trajectory of a length $M_l$ and a frame shift $\Delta n$, corresponding to a time shift $\delta t = \Delta n\Delta t$, is defined as:

$$\mu_{v,l}\left(\Delta n\right) = \frac{1}{M_l - \Delta n} \sum_{n=0}^{M_l - \Delta n - 1} |x_l\left(n + \Delta n\right) - x_l\left(n\right)|^v,$$

where $x_l$ is the position vector ($x'(n)$, $y'(n)$) on an individual trajectory $l$ at time $n\Delta t$. To quantify the particle motion these moments are calculated for $v$ = 0, 1, 2,…, 6 and $\Delta n$ = 1, …, $M_{l/3}$. For all the moments of displacement the scaling coefficients $\gamma_v$ are determined by a linear least squares regression to ln $\mu_v$ versus ln $\delta t$. The regular diffusion constant for $\nu = 2$ ($D_2$) was obtained from the $y_0$ intercept of the regression line such as $D_2 = 4^{-1} * e^{y_0}$. To calculate the fraction of immobile particles, we measured the $D_2$ values of fixed antibodies on glass acquired in the same imaging conditions as in the cells. We defined mobile particles as those whose $D_2$ values were above the 90 percentile of the $D_2$ values measured for the antibodies on glass (in our case 0.002 µm²/s). The slope of the plot of $\gamma_\nu$ versus $\nu$, called moment scale spectrum (MSS) was used to classify trajectories according to their motions in confined (<0.2), sub-diffusive (>0.2 to <0.5), and free (>0.5).

## Statistical analysis

All results were analyzed using GraphPad PRISM 7.0 (ns, p > R 0.05, *p % 0.05, **p % 0.001; ***p % 0.0001).

Paired *T*-test analysis was used in *Figures 1C, D, G, 2F, G, K, and 6G, E*.

Non-parametric Mann–Whitney tests were used in *Figure 3H,I*, *Figure 1—figure supplement 1K,L,M*, *Figure 3—figure supplement 1A,B, C, E, J, M*, *Figure 4—figure supplement 1B, C*, *Figure 6—figure supplement 1D*, *Figure 4B,C,D*, *Figure 6I*.

Kruskal–Wallis with Dunn's multiple comparison tests was used in *Figures 4F, G, J and 5E, F, H, I, J*.

Two-way analysis of variance with multiple comparison Bonferroni tests was used in *Figure 1B,H,J*, *Figure 2H,I,L*, *Figure 2—figure supplement 1C,E,I*, *Figure 3B, C, F, J, K*, *Figure 3—figure supplement 1F, G, H, K, L, N*, *Figure 5—figure supplement 1A, B, D*, *Figure 5B, C, D, G, K*, *Figure 6B-J*.

## Acknowledgements

The authors would like to thank C Martinez-Guillamon for initial experiments, J Andilla and M Marsal for technical support at the Super-resolution Light Nanoscopy (SLN) facility at ICFO. The research leading to these results has received funding from the European Commission H2020 Program (grant agreement ERC Adv788546 (NANO-MEMEC) (to MFG-P) and Marie Sklodowska-Curie grant 754558-PREBIST (to NM)), Government of Spain Severo Ochoa CEX2019-000910-S, State Research Agency (AEI) (PID2020-113068RB-I00/10.13039/501100011033 (to MFG-P), PID2019-109870RB-I00 (to JM-P), (PID2020-117405GB-100 to ML), RYC-2017-22227 (to FC), RYC-2015-17896 (to CM), and PID2019-106232RB-I00/10.13039/501100011033 (to FC)), Fundació CELLEX (Barcelona), Fundació Mir-Puig and the Generalitat de Catalunya through the CERCA program and AGAUR (Grant No. 2017SGR1000 to MFG-P). NI-U is supported by grant PID2020-117145RB-I00 from the Spanish Ministry of Science and Innovation. JM-P an NI-U are funded by the CIBER de Enfermedades Infecciosas.

## Additional information

### Competing interests

Felix Campelo: Reviewing editor, *eLife*. Nuria Izquierdo-Useros, Javier Martinez-Picado: declares patent applications related to the inhibition of viral interactions with Siglec-1. application reference EP23382072.9. The author declare no other competing interests. The other authors declare that no competing interests exist.

### Funding

| Funder | Grant reference number | Author |
|---|---|---|
| European Research Council | 788546 | Maria F Garcia-Parajo |
| European Commission | 754558 | Nicolas Mateos |
| Spanish National Plan for Scientific and Technical Research and Innovation | CEX2019-000910-S | Maria F Garcia-Parajo |
| Spanish National Plan for Scientific and Technical Research and Innovation | PID2020-113068RB-I00 / 10.13039/501100011033 | Maria F Garcia-Parajo |
| Spanish National Plan for Scientific and Technical Research and Innovation | PID2019-109870RB-I00 | Javier Martinez-Picado |
| Spanish National Plan for Scientific and Technical Research and Innovation | PID2020-117405GB-100 | Maier Lorizate |
| Spanish National Plan for Scientific and Technical Research and Innovation | RYC-2017-22227 | Felix Campelo |

| Funder | Grant reference number | Author |
|---|---|---|
| Spanish National Plan for Scientific and Technical Research and Innovation | RYC-2015-17896 | Carlo Manzo |
| Spanish National Plan for Scientific and Technical Research and Innovation | PID2019-106232RB-I00/10.13039/501100011033 | Felix Campelo |
| Spanish National Plan for Scientific and Technical Research and Innovation | PID2020-117145RB-I00 | Nuria Izquierdo-Useros |
| Generalitat de Catalunya | 2017SGR1000 | Maria F Garcia-Parajo |
| FUNDACIÓ Privada MIR-PUIG | | Maria F Garcia-Parajo |
| Fundación Cellex | | Maria F Garcia-Parajo |

The funders had no role in study design, data collection, and interpretation, or the decision to submit the work for publication.

## Author contributions

Enric Gutiérrez-Martínez, Conceptualization, Resources, Data curation, Formal analysis, Validation, Investigation, Visualization, Methodology, Writing – original draft, Project administration; Susana Benet Garrabé, Itziar Erkizia, Resources, Investigation, Methodology; Nicolas Mateos, Resources, Software, Formal analysis; Jon Ander Nieto-Garai, Resources, Writing – original draft; Maier Lorizate, Resources, Methodology, Writing – original draft; Kyra JE Borgman, Investigation, Methodology; Carlo Manzo, Resources, Software, Methodology; Felix Campelo, Data curation, Investigation, Methodology, Writing – original draft; Nuria Izquierdo-Useros, Javier Martinez-Picado, Conceptualization, Supervision, Writing – original draft, Writing – review and editing; Maria F Garcia-Parajo, Conceptualization, Supervision, Funding acquisition, Investigation, Writing – original draft, Project administration, Writing – review and editing

## Author ORCIDs

Jon Ander Nieto-Garai ![ORCID] http://orcid.org/0000-0002-8665-099X
Kyra JE Borgman ![ORCID] http://orcid.org/0000-0001-7898-2911
Carlo Manzo ![ORCID] http://orcid.org/0000-0002-8625-0996
Felix Campelo ![ORCID] http://orcid.org/0000-0002-0786-9548
Nuria Izquierdo-Useros ![ORCID] http://orcid.org/0000-0002-1039-1821
Javier Martinez-Picado ![ORCID] http://orcid.org/0000-0002-4916-2129
Maria F Garcia-Parajo ![ORCID] http://orcid.org/0000-0001-6618-3944

## Decision letter and Author response

Decision letter https://doi.org/10.7554/eLife.78836.sa1
Author response https://doi.org/10.7554/eLife.78836.sa2

# Additional files

## Supplementary files

• MDAR checklist

## Data availability

All data generated or analyzed during this study are included in the manuscript and supporting file; Source Data files have been provided for Figures 1–6 and for the corresponding figure supplements.

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
