## [Editor Report]

Siglec-1 (CD169), a plasma membrane-associated sialic acid-binding lectin, has been implicated in the capture of HIV and other viruses by dendritic cells and macrophages, however, the molecular details of how HIV particles are captured by Siglec-1 are poorly understood. In this important paper, the authors use super-resolution imaging methods to analyse the cell surface distribution of Siglec-1 on immature and mature dendritic cells to study the regulation of Siglec-1 distribution by actin and regulators of actin polymerization and to understand how virus-Siglec-1 engagement leads to virus sequestration within so-called virus containing compartments. This compelling study has relevance for researchers studying the engagement of HIV and many other viruses with cells, as well as researchers interested in the mechanisms regulating receptor distribution and function on cells.

---

## [Decision Letter]

**Decision letter after peer review:**

Thank you for submitting your article "Actin-regulated Siglec-1 nanoclustering influences HIV-1 capture and virus-containing compartment formation in dendritic cells" for consideration by *eLife*. Your article has been reviewed by 3 peer reviewers, one of whom is a member of our Board of Reviewing Editors, and the evaluation has been overseen by a Reviewing Editor and Suzanne Pfeffer as the Senior Editor. The following individuals involved in the review of your submission have agreed to reveal their identity: Walther Mothes (Reviewer #2); Paul Spearman (Reviewer #3).

Essential revisions:

1) Insufficient experimental detail is provided for some aspects of the paper. In particular, the information for the experiments presented in Figure 1 is currently inadequate. Please provide information on the antibodies used.

2) Antibodies are dimers and can cluster Siglec-1. It is important that single chains or Fabs are used. The preparation of monochain anti-Siglec-1 antibodies and Fab-Atto conjugates is described in the methods, but their experimental use does not appear in the results. Please present data using these monovalent reagents compared to the intact antibody and double labelling methods.

3) Please provide more information on what is referred to as 'ring-shaped compartment'. (Line 373). Is the observed labelling on the cell surface or internal? Have these structures been imaged in 3D? While I don't think it necessary to provide further details on the VCC, it is currently unclear what this pattern of staining represents.

4) Please consider and address all the comments raised by the reviewers either through the provision of additional experimental data or modifications to the text of the manuscript.

*Reviewer #1 (Recommendations for the authors):*

One fundamental problem with imaging receptors in this way is knowing whether single spots correspond to individual or multiple receptor molecules. In Figure 1 the authors have gone to some lengths to understand what the labelling patterns show quantitatively. How confident can they be that the data in Figure 1B really represent 1, 2, 3, or 4 Siglec molecules? Do Siglecs form oligomers? Using two layers of antibody, as seems to be the case here, and PFA fixation, is there any possibility for antibodies to affect the distribution of Siglec-1? Such antibody-induced perturbations might be enhanced by increased Siglec-1 expression, rather than changes in the activation status of the cell.

In Materials and methods, the authors describe making monovalent anti-Siglec-1 Abs. Was Hsn 7D2 used, if so, this should be clearly stated? A gel showing the preparation should be shown in the suppl. data. I could not see where these monovalent Abs were used in the study.

Line 189, the authors claim 'the association of Siglec-1 with high-density regions of cortical actin …… was not affected by Arp2/3 inhibition' – although analysis of just the cortical regions of the cell may suggest little effect of CK666, Siglec-1 labelling over the middle of the cell suggests there is an effect of this inhibitor (Figure 2C). Moreover, it seems the ROI's indicated in these images differ; the top two being over actin-rich areas and the bottom SMIFH2 region being over an actin-depleted area. How were ROIs selected? Do the authors have controls to ensure CK666 and SMIFH2 had the expected effects on ARP2/3, formins, and actin?

Line 197: in this paragraph, the authors verify the effect of SMIFH2 on Siglec-1 using SPT; did they also do SPT with CK666 treated cells?

The methods for imaging and image analysis are presented in detail, but there are some aspects of the methods that need to be better explained. E.g., Line 547 refers to 'viral stocks' but the methods describe only the production of VLPs – these are not viruses. The text in this section and the relevant Results section (pages 10/11) need to be changed to avoid suggesting that an infectious virus was used. The VLPs were made using Gag and Env plasmids. Do the authors know if Env is present in the VLPs? Alternatively, do they know whether the results observed with VLPs are influenced at all by Env? Did they use VLPs lacking Env and, if so, what were the results? If Env is incorporated into the VLPs it is likely to be organised differently from that on mature virions as the lack of protease activity in the VLPs would prevent maturation-associated changes in Env distribution.

Can the authors indicate how the concentrations of GM-1 used in LUVs compare to those of the VLPs used in these experiments?

Line 373: can the authors be clearer about what they call a 'ring-shaped compartment'. Is the observed labelling on the cell surface or internal? Have these structures been imaged in 3D?

*Reviewer #2 (Recommendations for the authors):*

Insufficient experimental detail is provided for Figure 1. What anti-Siglec-1 antibodies, and what secondary antibodies were used? Antibodies are dimers and can cluster Siglec-1. It is important that single chains or Fabs are used. Monochain anti-Siglec-1 antibodies and Fab-Atto conjugates are listed in the methods, but the actual experimental conditions for Figure 1 are neither mentioned in the text nor in the figure legend.

How is the analysis of actin affected by changes in cell shape, which are dramatic in the transition from iDCs to mDCs?

The paper is narrowly focused on DCs but macrophages express CD169 and trans-infected permissive cells, particularly in vivo and mechanistic studies relevant to the current study like the formation of virus containing vacuoles in a Siglec-1 dependent manner should be cited and discussed, e.g. PMID: 28129379.

*Reviewer #3 (Recommendations for the authors):*

The only direct suggestion for this work is to better define the VCC in the context of the experiments shown, using other VCC markers and indicating the cellular position of the VCC (ie internal or merely concentrated virions on the surface). I don't think you can call something a VCC when it is concentrated VLP + Siglec clustering that has not undergone invagination/internalization and is characterized also by typical VCC markers.

Some biological significance to trans-infection events could be easily added to this report, showing that disruption of formin or RhoA pathways prevents trans-infection. This is not really necessary in my opinion for the current report but would boost overall significance.

I would like to see additional comments on the mechanism and significance of Siglec-1 capture and movement on the apical vs basal cell surface. The manuscript reads as if VLPs are all concentrated on the basal surface, yet VLPs are often captured on the exposed apical surface by Siglec-1. For the images shown, especially where there is a ring of Siglec or Siglec and VLPs, it would help to show a z-series or 3D view to see where this ring is. Is it around the cell soma but on the apical surface, or really on the basal surface?

---

## [Author Response]

Reviewer #1 (Recommendations for the authors):One fundamental problem with imaging receptors in this way is knowing whether single spots correspond to individual or multiple receptor molecules. In Figure 1 the authors have gone to some lengths to understand what the labelling patterns show quantitatively. How confident can they be that the data in Figure 1B really represent 1, 2, 3, or 4 Siglec molecules? Do Siglecs form oligomers? Using two layers of antibody, as seems to be the case here, and PFA fixation, is there any possibility for antibodies to affect the distribution of Siglec-1? Such antibody-induced perturbations might be enhanced by increased Siglec-1 expression, rather than changes in the activation status of the cell.

We appreciate the concern of the reviewer. Indeed the experiments shown in Figure 1 have been performed using secondary antibody labeling which is the standard for super-resolution STED imaging. Obviously, labeling has been performed after fixing the cells with 4% PFA which has been shown by multiple groups (including ours) to fully immobilize receptors on the cell membrane (Leyton-Puig et al., *Methods and Tech.* 2016; Bakker et al., *PNAS* 2012; Torreno-Pina et al., *PNAS* 2014). In these conditions, the possibility of artefactual clustering is negligible. Having said that, we agree with the reviewer that in conditions where the expression level of the receptors is very high, potential Ab-induced perturbations might still occur. Thus, to fully rule out potential artifacts due to Siglec-1 labeling or fixation by PFA, we performed two different control experiments; (1) we used single-chain antibodies (Abs.) for labeling Siglec-1 in cells fixed with 4% PFA (as used in our original submission) and, (2) we use single-chain Abs. on cells fixed with 1% PFA + 0.2% GA (glutaraldehyde). Analysis on multiple STED images in these two types of control experiments showed no difference in terms of nanoclustering capacity of Siglec-1 on activated DCs as compared to the full Ab labeling, regardless of the fixation protocol. Thus, these controls rule out any potential artefacts due to labeling conditions or sample fixation. We have now included these control experiments as Figure 1—figure supplement 1E-I, Figure 1—figure supplement 1-source data 1 and Figure 1—figure supplement 1-source data 2 and added a paragraph in the revised manuscript regarding these controls and results (see page 5-6).

In Materials and methods, the authors describe making monovalent anti-Siglec-1 Abs. Was Hsn 7D2 used, if so, this should be clearly stated? A gel showing the preparation should be shown in the suppl. data. I could not see where these monovalent Abs were used in the study.

We apologize for omitting this information in the original manuscript. Single-chain Abs were used for all the single molecule live imaging experiments (Figure 1E-J; Figure 2J-L). Since direct fluorophore labeling of the single chains was very inefficient and resulted in weak signals, we used Fab-Atto488 anti-mouse to label the single-chains Abs. And yes, we used a mouse monoclonal antibody to Siglec-1 (Hsn 7D2), obtained from Abcam. We have now included all the relevant information at the appropriate parts in the manuscript as well as included the gel data on the full and single chains (see Figure 1—figure supplement 1E and Figure 1—figure supplement 1-source data 1).

Line 189, the authors claim 'the association of Siglec-1 with high-density regions of cortical actin …… was not affected by Arp2/3 inhibition' – although analysis of just the cortical regions of the cell may suggest little effect of CK666, Siglec-1 labelling over the middle of the cell suggests there is an effect of this inhibitor (Figure 2C). Moreover, it seems the ROI's indicated in these images differ; the top two being over actin-rich areas and the bottom SMIFH2 region being over an actin-depleted area. How were ROIs selected? Do the authors have controls to ensure CK666 and SMIFH2 had the expected effects on ARP2/3, formins, and actin?

Although the representative images shown in Figure 2C might indeed suggest there is also an effect of CK666, our observations are based on the statistical analysis over multiple and random ROIs per cell, and performed on at least 10 cells per condition (control, CK666 treated and SMIFH2 treated) and for different donors. As such, the analysis is extremely robust. To avoid any confusion to the reader, we have now slightly modified the lowest panel of Figure 2C and provide an enlarged view on an actin-rich region, similar to the control and CK666 treated samples. Although we did not perform specific controls of CK666 and SMIFH2 regarding the expected effects on Arp2/3 and formins respectively, we and many others in the field have routinely used these treatments to perturb specific components of actin polymerization (Henson et al. *Mol. Biol. Cell* 2015, Freeman et al. *Cell* 2018, Sil et al. *Mol. Biol. Cell* 2019; Nishimura et al. *J. Cell Sci.* 2021, amongst others).

Line 197: in this paragraph, the authors verify the effect of SMIFH2 on Siglec-1 using SPT; did they also do SPT with CK666 treated cells?

SPT experiments including thorough data analysis is quite demanding and laborious. Therefore, considering the large amount of different experiments reported in the manuscript and the fact that we did not observe statistical differences upon CK666 treatment on the organization of Siglec-1 by means of STED, we exclusively focused on SPT experiments and their analysis upon SMIFH2 perturbation. Although we could perform SPT experiments on CK666 treated cells, we believe that they would not provide additional insights. Instead, during the course of our research we considered more relevant to study the relationship between formins and Siglec-1 nanoscale organization in a physiological situation in which different types of actin pools simultaneously occur at specific regions of the plasma membrane. As shown and quantified in the *Figure 2—figure supplement 1F-I* and *Figure 2—figure supplement 1-source data 1*, Siglec-1 was remarkably enriched at the cell rear, where it is known that actin polarization is mainly formin-dependent (Lämmermann and Sixt, *Curr Opin Cell Biol*. 2009; Paluch et al *Annu Rev Cell Dev Biol*. 2016; Vargas et al. *Nat Cell Biol.* 2016). Moreover, a significant enhancement of Siglec-1 nanoclusters was detected at the cell rear as compared to the lamellipodium at the front which is mainly dependent on Arp2/3 branched actin.

The methods for imaging and image analysis are presented in detail, but there are some aspects of the methods that need to be better explained. E.g., Line 547 refers to 'viral stocks' but the methods describe only the production of VLPs – these are not viruses. The text in this section and the relevant Results section (pages 10/11) need to be changed to avoid suggesting that an infectious virus was used. The VLPs were made using Gag and Env plasmids. Do the authors know if Env is present in the VLPs? Alternatively, do they know whether the results observed with VLPs are influenced at all by Env? Did they use VLPs lacking Env and, if so, what were the results? If Env is incorporated into the VLPs it is likely to be organised differently from that on mature virions as the lack of protease activity in the VLPs would prevent maturation-associated changes in Env distribution.

We apologize for the confusion. The methods have now been changed to “Viral like particle stock production” and also in the indicated pages.

Regarding the presence of Env in the VLPs: the co-transfection of HIV-1 Gag plasmids along with an HIV-1 envelope plasmid to produce a pseudotyped VLP is well established in the literature. Although we have not directly checked for the presence of the envelope glycoproteins in these VLPs, we have used the same methodology to pseudotype a single-cycle infectious HIV-1-δ-envelope construct that contains a luciferase reporter gene. These single cycle HIV-1 pseudotyped viral particles effectively infect target cells, clearly showing that envelopes are incorporated into these particles allowing for productive infection (Izquierdo-Useros et al. *J. Virol* 2007).

Regarding the potential influence of Env: We and others have extensively described that the envelope glycoprotein of HIV-1 is not required for the viral recognition mediated by Siglec-1 (Izquierdo-Useros et al. *PloS Biol* 2012; Puryear et al. *PNAS* 2012 and *PLoS Pathog* 2013) as we also confirm in this manuscript using LUVs lacking this molecule. Indeed, prior data from our group already showed how HIV-1 VLPs lacking the envelope glycoprotein and HIV-1wt viruses accumulate within the same virus containing compartment in mature dendritic cells (Izquierdo-Useros et al. *Blood* 2009). That is also the reason why this receptor is able to interact with many different enveloped viruses that display very distinct envelope glycoproteins, such as Ebola Virus, Marburg Virus, or even SARS-CoV-2, that contain sialylated gangliosides (Perez-Zsolt et al. *Nat Microbiol* 2019 and *Cell Mol Immunol* 2021). In the particular case of HIV-1, it has been shown that only 14 +/- 7 envelope trimers are incorporated per virion (Zhu et al. *Nature* 2006). This incorporation into the VLPs is not dependent on the protease activity, which acts internally inside the viral particle to cleave gag polyproteins and release matrix, capsid, nucleocapsid and p6 subunits. However, as the reviewer mentions, in mature HIV-1 particles, the envelope glycoprotein trimers are able to diffuse and cluster together, favoring the infectious process of the virus, which is not the focus of our study.

Can the authors indicate how the concentrations of GM-1 used in LUVs compare to those of the VLPs used in these experiments?

Lipidomic analysis of HIV-1 membranes estimated that there are 300,000 lipids per viral particle and 12,000 GM3 molecules per virion (Brügger et al. *PNAS* 2006; Chan et al *J Virol* 2008). Other gangliosides such as GM1 are also present in HIV-1 membranes (Izquierdo-Useros et al. *PloS Biol* 2012) but they remain to be accurately quantified. Taking into consideration only those calculations that have been accurately performed, 12,000 gangliosides constitute approximately 4% of the total lipid molecules present in one virus. That is in fact the reason why we have used a 4 % of ganglioside content to mimic the lipid composition of HIV-1 or VLPs. The presence of thousands of sialyllactose-containing gangliosides in the viral membrane is expected to support high-avidity interactions with clusters of Siglec1 receptors, as we show here by decreasing the percentage of ganglioside content in the LUVs used, which alters LUV recognition. We have now added this information in the Materials and Methods section and include additional references regarding the number of gangliosides in the viral membrane (see page 21).

Line 373: can the authors be clearer about what they call a 'ring-shaped compartment'. Is the observed labelling on the cell surface or internal? Have these structures been imaged in 3D?

By “ring-shaped compartment” we refer to the area of the cell a few microns above the basal membrane being physically constrained by the enrichment of actin, Siglec-1 and VLPs (as shown in Figure 6A at 30 min of VLPs or 4% GM-1 LUVs incubation). The observed labeling of Siglec-1 and VLPs (or LUVs) is on the surface of the cells (not internal) imaged by 3D confocal imaging at different z positions. The images were obtained on fixed cells labeled for actin, Siglec-1 and VLPs (or LUVs), and by performing 3D stacks by confocal microscopy. As explained in the Material and Methods section the whole surface of cells was covered in stacks of 10-15 images from the bottom to the top. We applied intensity thresholds for each staining and calculated the mean intensity values in axial planes of 1.5 µm. To measure the area of cells at different axial planes after VLP/LUV capture, we delimited the edges of cells stained with SiRactin applying a binary mask for each plane and quantified the area enclosed in such perimeter. We have now clarified better the different panels of Figure 6 where the ring-shaped regions are shown.

Reviewer #2 (Recommendations for the authors):Insufficient experimental detail is provided for Figure 1. What anti-Siglec-1 antibodies, and what secondary antibodies were used? Antibodies are dimers and can cluster Siglec-1. It is important that single chains or Fabs are used. Monochain anti-Siglec-1 antibodies and Fab-Atto conjugates are listed in the methods, but the actual experimental conditions for Figure 1 are neither mentioned in the text nor in the figure legend.

We apologize for the lack of clarity regarding the labeling strategies used for the different imaging experiments reported in the manuscript. For the data shown in Figure 1A-D we used primary and secondary Abs. and performed labeling after fixing the cells, as it is the standard method for STED imaging. Still, we acknowledge that in conditions where the expression of receptors is high, such as in the case of Siglec-1 in mDCs, this labeling strategy might introduce potential perturbations on the organization of the receptors. Therefore, and following also the request of reviewer 1, we performed control experiments by labeling the cells using single-chain Abs. and under different fixation conditions. These new results showed no difference in terms of nanoclustering capacity of Siglec-1 on activated DCs as compared to the full Ab labeling and regardless of the fixation method. Overall, these controls rule out any potential artefacts due to labeling conditions or sample fixation. The results are now included in *Figure 1—figure supplement 1E-I* together with the corresponding source data, and mentioned in the main text of the manuscript. In addition, we also specify now in the caption of Figure 1 how the labeling of Siglec-1 for STED imaging has been performed. In addition, in the case of the single molecule imaging data obtained in living cells and shown in Figures 1E-J and Figures 2J-L, we have used single-chain Abs. labeled with Fab-Atto488. We have also included this information in the caption of Figure 1.

How is the analysis of actin affected by changes in cell shape, which are dramatic in the transition from iDCs to mDCs?

In fact, we did not perform a detail analysis on actin re-organization since as the reviewer mentions, there are major changes in cell shape in the transition from iDCs to mDCs. Instead, we focused our studies on assessing the location on the plasma membrane where siglec-1 confinement and nanoclustering was more prominent, in relation to the intensity of the actin signal.

The paper is narrowly focused on DCs but macrophages express CD169 and trans-infected permissive cells, particularly in vivo and mechanistic studies relevant to the current study like the formation of virus containing vacuoles in a Siglec-1 dependent manner should be cited and discussed, e.g. PMID: 28129379.

We thank the reviewer for this suggestion and have now incorporated a small paragraph in the discussion as well as providing the reference (see page 19).

Reviewer #3 (Recommendations for the authors):The only direct suggestion for this work is to better define the VCC in the context of the experiments shown, using other VCC markers and indicating the cellular position of the VCC (ie internal or merely concentrated virions on the surface). I don't think you can call something a VCC when it is concentrated VLP + Siglec clustering that has not undergone invagination/internalization and is characterized also by typical VCC markers.

As mentioned above, our study provides important insights into the earliest events that later lead to VCC formation. Thus, we do not focus on the actual VCC in the current manuscript since it has been extensively studied by others (and us) in the past. This is the reason why we always refer to a “ring-shaped compartment”, i.e., the area of the cell above the basal membrane being physically constrained by the enrichment of actin, Siglec-1 and VLPs (as shown in Figure 6A at 30 min of VLPs or 4% GM-1-LUVs incubation), rather than to the VCC. We believe that the formation of this ring-compartment precedes the final formation of the VCC. To avoid any misunderstanding in the manuscript we have now clarified the sentences that relate to the VCC, indicating that the formation of the VCC will eventually occur at later stages.

Some biological significance to trans-infection events could be easily added to this report, showing that disruption of formin or RhoA pathways prevents trans-infection. This is not really necessary in my opinion for the current report but would boost overall significance.I would like to see additional comments on the mechanism and significance of Siglec-1 capture and movement on the apical vs basal cell surface. The manuscript reads as if VLPs are all concentrated on the basal surface, yet VLPs are often captured on the exposed apical surface by Siglec-1. For the images shown, especially where there is a ring of Siglec or Siglec and VLPs, it would help to show a z-series or 3D view to see where this ring is. Is it around the cell soma but on the apical surface, or really on the basal surface?

We apologize for the confusion as indeed the ring is not formed at the basal surface but a few microns above it. To clarify: the images shown in Figure 6A correspond to intensity projections of 3D-confocal images taken at every 1.5 µm from the basal to the apical membrane (between 10-15 stacks per image). On the bottom part of Figure 6A we show the outlined perimeter obtained from each stack, color coded according to the axial position of each plane with respect to the basal membrane. Visually, the highest constriction occurs at a plane around 4-6 µm above the basal surface. This is robustly quantified in *Figure 6—figure supplement 1A,B* and associated source data where we show that the highest colocalization between Siglec-1, actin and VLPs (or LUVs) occurs at axial distances around 4-6 µm *above* the basal membrane. We have now clarified the main text and the caption of Figure 6 to indicate better where the region of highest constriction and recruitment of Siglec-1, actin and VLPs (or LUVs) occurs.